# AC-DiT: Adaptive Coordination Diffusion Transformer for Mobile Manipulation

**Sixiang Chen**[1,4]*, **Jiaming Liu**[1]*†, **Siyuan Qian**[1,4]*, **Han Jiang**[2], **Xiaoqi Li**[1], **Renrui Zhang**[3],
**Zhuoyang Liu**[1], **Chenyang Gu**[1,4], **Chengkai Hou**[1], **Pengwei Wang**[4],
**Zhongyuan Wang**[4], **Shanghang Zhang**[1,4] ✉

[1]State Key Laboratory of Multimedia Information Processing, School of Computer Science, Peking University;
[2]Nanjing University; [3]CUHK; [4]Beijing Academy of Artificial Intelligence (BAAI)
**Project web page:** ac-dit.github.io

## Abstract

Recently, mobile manipulation has attracted increasing attention for enabling language-conditioned robotic control in household tasks. However, existing methods still face challenges in coordinating mobile base and manipulator, primarily due to two limitations. On the one hand, they fail to explicitly model the influence of the mobile base on manipulator control, which easily leads to error accumulation under high degrees of freedom. On the other hand, they treat the entire mobile manipulation process with the same visual observation modality (e.g., either all 2D or all 3D), overlooking the distinct multimodal perception requirements at different stages during mobile manipulation. To address this, we propose the Adaptive Coordination Diffusion Transformer (AC-DiT), which enhances mobile base and manipulator coordination for end-to-end mobile manipulation. First, since the motion of the mobile base directly influences the manipulator's actions, we introduce a mobility-to-body conditioning mechanism that guides the model to first extract base motion representations, which are then used as context prior for predicting whole-body actions. This enables whole-body control that accounts for the potential impact of the mobile base's motion. Second, to meet the perception requirements at different stages of mobile manipulation, we design a perception-aware multimodal conditioning strategy that dynamically adjusts the fusion weights between various 2D visual images and 3D point clouds, yielding visual features tailored to the current perceptual needs. This allows the model to, for example, adaptively rely more on 2D inputs when semantic information is crucial for action prediction, while placing greater emphasis on 3D geometric information when precise spatial understanding is required. We empirically validate AC-DiT through extensive experiments on both simulated and real-world mobile manipulation tasks, demonstrating superior performance compared to existing methods.

## 1 Introduction

Mobile manipulation, which integrates mobile platforms with manipulative capabilities to perform complex household tasks under natural language conditions, has gained substantial attention. Some two-stage approaches [1, 2, 3, 4, 5] leverage task planners (e.g., VLMs or LLMs) to decompose long-horizon tasks into sub-tasks, which are then executed sequentially by atomic skill modules. In contrast, end-to-end methods [6, 7, 8, 9, 10] typically employ imitation learning or reinforcement learning to jointly optimize the control of both the mobile base and the manipulator. However, these

---

*Equal contribution, †Project lead, ✉Corresponding author.

39th Conference on Neural Information Processing Systems (NeurIPS 2025).

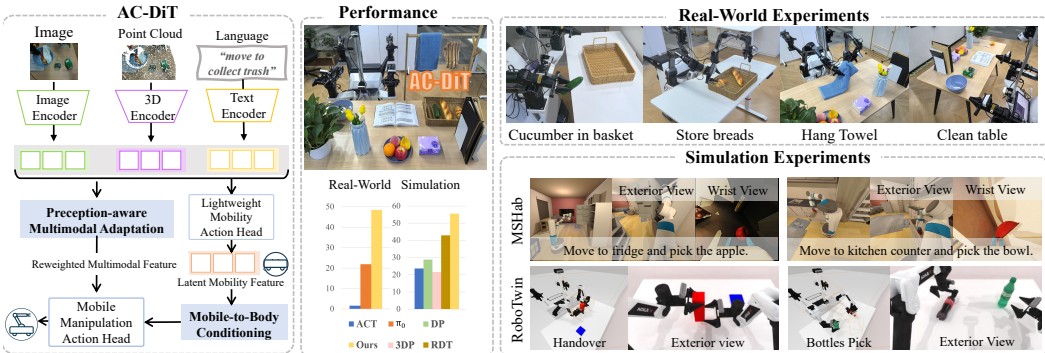

Figure 1: **Overview of AC-DiT.** The proposed end-to-end mobile manipulation framework enhances the coordination between the mobile base and the manipulator by introducing two key mechanisms: mobile-to-body conditioning and perception-aware multimodal adaptation. The former enables action prediction that conditioning on how upcoming mobile base movements may affect manipulator control, thereby reducing error accumulation. The latter constructs multimodal features tailored to the perception requirements at different stages of the mobile manipulation process. Under this paradigm, AC-DiT demonstrates superior performance in both simulation and real-world environments.

approaches still face significant challenges in effectively coordinating the two components. One key limitation is the lack of explicit modeling of the coordination relationship between the mobile base and manipulator control, which is crucial because even minor errors in the mobile base's movement can severely impact the manipulator's performance. In addition, they use the same visual observation modality throughout the entire mobile manipulation process, such as consistently relying on either 2D images or point clouds as input, which overlooks the distinct visual perception requirements of each stage. For example, during object localization, semantically rich 2D images enable better scene understanding and target identification, while during interaction, geometrically detailed point clouds are essential for ensuring precise manipulation. Building on these challenges, we ask: "*Can we develop an end-to-end mobile manipulation framework that effectively leverages the relationship between the mobile base and the manipulator, as well as the distinct characteristics of different task stages, to achieve accurate coordination?*"

To solve the above challenge, as shown in Figure 1, we propose the Adaptive Coordination Diffusion Transformer (AC-DiT), an end-to-end diffusion transformer (DiT) model for mobile manipulation. First, to better formulate the relationship of mobile base while mitigating potential error accumulation from the base's motion that could affect manipulator control, we introduce **a mobility-to-body condition mechanism** to enhance whole-body coordination. Specifically, we attach a lightweight mobility policy head (e.g., DiT [11]) to the encoders and pretrain it using only mobile base actions, enabling it to extract latent mobility features that capture base motion representation. These features are then used as conditional inputs in the subsequent training of the mobile manipulation action head, guiding the prediction of mobile manipulation actions. By providing prior information about the mobile base's movement, the model can better plan its actions in response to the expected movement of the base, thereby improving the overall coordination and avoiding error accumulation.

Second, to adapt the perception system to the varying demands of different stages in the mobile manipulation process, we propose a **perception-aware multimodal adaptation mechanism** that dynamically assigns appropriate weights to different visual modalities. Specifically, given extracted language features, image features from multiple views, and point cloud features, we first project them into a shared feature space using a lightweight projector. We then compute the cosine similarity between each visual modality (image and point cloud) and the language features to estimate their importance. These similarity-based weights are applied to the original visual tokens and used to produce perception-aware visual representations, which are then fed into policy head for action prediction. In this way, we empirically discover that AC-DiT can adaptively adjust its reliance on different modalities based on stage context—for example, relying more on 2D inputs when semantic information is more critical, and focusing more on 3D inputs when geometric precision is required. Moreover, it can actively downweight the influence of uninformative views, such as images that capture only floors or walls.

To comprehensively evaluate AC-DiT, as shown in Figure 1, we conduct extensive experiments across multiple tasks, including both simulated environments (e.g., ManiSkill-HAB [12]) and real-world scenarios. We compare our method with several imitation learning (e.g., DP [13], DP3 [14], and RDT [11]) and mobile manipulation approaches (e.g., ACT [7, 6] and $\pi_0$ [15]), and our method demonstrates a significant advantage over others. In summary, our contributions are as follows:

- We introduce the Adaptive Coordination Diffusion Transformer (AC-DiT), an end-to-end framework designed to enhance coordination for mobile manipulation by explicitly modeling the relationship between the mobile base and the manipulator, and by adapting to the distinct visual perception requirements at different mobile manipulation stages.

- To achieve this, we propose a *mobility-to-body conditioning mechanism* that conditions overall mobile manipulation actions on latent mobility representation, enhancing coordination and reducing cumulative errors. We also design a *perception-aware multimodal adaptation mechanism* that adaptively assigns importance weights to different visual inputs to satisfy the perception requirements of each stage in the mobile manipulation process.

- Experiments demonstrate that our method outperforms state-of-the-art baselines in both simulated and real-world mobile manipulation tasks, highlighting its enhanced coordination and robust action generation capabilities.

## 2 Related works

**Mobile Manipulation.** Mobile manipulation tasks require robots to coordinate the motions of a mobile base and a manipulator arm in response to human instructions. A common approach [1, 2, 3, 4, 5] leverages a high-level planner [16, 17], typically a Vision-Language Model (VLM), to decompose tasks into subtasks, which are then sequentially executed by low-level atomic skill modules (e.g., imitation learning [18, 6, 7, 19, 20], reinforcement learning [21, 22, 23, 24, 25], or foundation models [26, 27]). Alternatively, end-to-end frameworks jointly model the degrees of freedom of both the base and the manipulator within a unified action space, using reinforcement learning [28, 29, 30] or imitation learning [6, 7, 8, 9, 10] to enable holistic optimization for scenarios requiring concurrent or rapidly alternating locomotion and manipulation. However, existing methods rely on single-modality perception (either 2D or 3D) and overlook the modality-specific perceptual granularity required by mobile manipulation. Moreover, they fail to explicitly capture the kinematic and dynamic interdependencies between locomotion and manipulation. Therefore, we propose AC-DiT to address these critical gaps in an end-to-end manner.

**Vision-Language-Action (VLA) models.** Visual-Language-Action (VLA) models empower large-scale models to interpret visual and linguistic inputs while generating corresponding robot actions [31, 32, 33, 34]. Representative works such as OpenVLA [35], RT-2 [36], and ManipLLM [37] discretize the continuous action space into bins and adopt next-token prediction for action generation, which inevitably compromises the natural continuity of robotic motion. To mitigate this, recent methods replace the VLM's tokenizer with continuous-valued regression heads [38, 39, 40, 41, 42] (e.g., MLPs, LSTMs), or integrate diffusion-based transformer modules [13, 43, 14, 44, 11, 45, 15, 46, 47, 48] to better capture the multimodal distribution of robot trajectories. Despite these advancements, existing VLA frameworks remain primarily limited to tabletop manipulation tasks, leaving the complex challenges of mobile manipulation largely unexplored. To address this gap, we propose the AC-DiT architecture, which adopts a diffusion-based action generation paradigm and is specifically designed to meet the distinct perceptual demands of different stages in mobile manipulation, while explicitly modeling the interdependencies between the mobile base and the manipulator.

## 3 Methods

In Section 3.1 and 3.2, we begin with the problem formulation and overall architecture of our proposed AC-DiT framework. In Section 3.3 and Section 3.4, we provide a detailed introduction to the mobility-to-body conditioning mechanism and perception-aware multimodal adaptation mechanism, respectively. We then demonstrate training objectives in Section 3.5

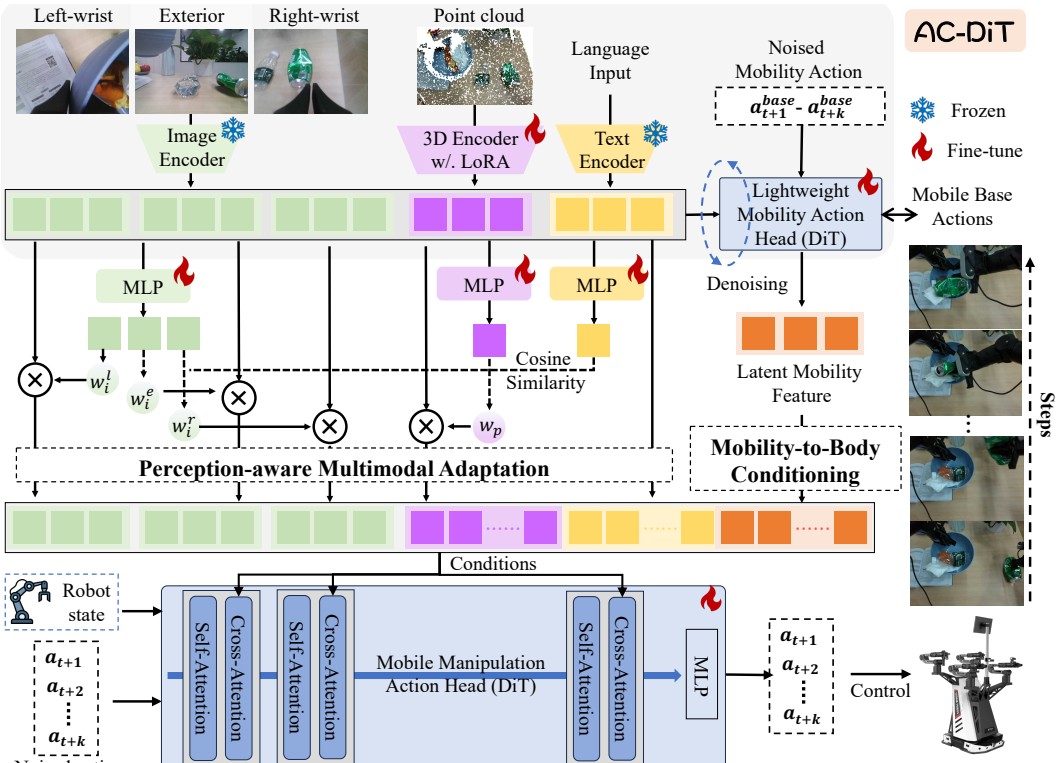

Figure 2: **AC-DiT framework**. We first train the modules in the grey-shaded region under the supervision of mobile base actions, allowing the lightweight mobility action head to learn to extract latent mobility features. After this, we optimize the entire AC-DiT model, enabling the mobile manipulation action head to predict both mobile base and manipulator actions. With the *Mobility-to-Body Conditioning* mechanism, this action head conditions on the extracted latent mobility features, allowing whole-body action prediction to account for the influence of mobile base motion. Meanwhile, the *Perception-Aware Multimodal Adaptation* mechanism enables this action head to adaptively assign different importance weights to various visual input features, resulting in a perception-aware visual condition tailored to the perception needs of different manipulation stages.

## 3.1 Problem Formulation

We model mobile manipulation as a temporal sequence prediction problem, where the agent processes a history of observations $\mathbf{o}_{t-\tau:t} = \{\mathbf{o}_{t-\tau}, \ldots, \mathbf{o}_t\}$ and language condition $\ell$ to predict a future action sequence $\mathbf{a}_{t+1:t+k} = \{\mathbf{a}_{t+1}, \ldots, \mathbf{a}_{t+k}\}$. The **observation** $\mathbf{o}_t$ at each timestamp $t$ is represented as $\mathbf{o}_t = (V_{2D}^t, V_{3D}^t, z_t, c)$, where $V_{2D}^t$ contains images from exterior, left-wrist, and right-wrist cameras, $V_{3D}^t$ represents the point cloud captured at timestamp $t$, $z_t$ is the robot state including the linear velocity and the angular velocity of the mobile base and joint positions of the manipulator, and $c$ is the control frequency of the robot. Meanwhile, we define coordination in mobile manipulation as an end-to-end control paradigm where a multi-component robot (in our case, mobile base and arm system) simultaneously generates synchronized action decisions for all components at each time step [7, 15], rather than employing sequential or phase-based control strategies. Therefore, the predicted **action chunk** $\mathbf{a}_{t+1:t+k}$ includes the mobile base control $\mathbf{a}_{t+1:t+k}^{\text{base}}$ (linear/angular velocities) and the manipulator control $\mathbf{a}_{t+1:t+k}^{\text{manipulator}}$ (joint positions). The goal is to learn an end-to-end policy $\pi(\mathbf{a}_{t+1:t+k}|\mathbf{o}_{t-\tau:t}, \ell)$, which is trained on dataset $\mathcal{D} = \{(\ell^{(i)}, \mathbf{o}_t^{(i)}, \mathbf{a}_t^{(i)}) \mid 0 \le t < T^{(i)}, 1 \le i \le N\}$ of $N$ trajectories, where $T^{(i)}$ is the length of the $i$-th trajectory.

## 3.2 Overall Architecture

As shown in Figure 2, AC-DiT comprises an image encoder, a 3D encoder, and a text encoder to extract features from 2D images of three camera views, 3D point cloud, and language input,

respectively. To enable a unified feature representation across modalities, we adopt SigLIP [49] as the backbone for all three encoders. Specifically, the SigLIP image encoder is used to extract exterior view feature $\mathbf{F}_{2D}^f$, left-wrist view feature $\mathbf{F}_{2D}^l$, right-wrist view feature $\mathbf{F}_{2D}^r$, which are then concatenated as 2D visual features $\mathbf{F}_{2D}$. For the point cloud input, we follow the strategy introduced in Lift3D [50], which transforms point clouds into token sequences compatible with 2D foundation models (e.g., SigLIP) with LoRA adapter [51]. This is achieved using a 3D tokenizer that processes the raw point cloud and aligns each 3D point with multiple 2D positional embeddings derived from several virtual planes projected from the 3D coordinates. The resulting 3D tokens, enriched with positional information, are then fed into the SigLIP image encoder to extract 3D geometric features $\mathbf{F}_{3D}$. For language input, we utilize the SigLIP text encoder to extract language features $\mathbf{F}_\ell$.

AC-DiT incorporates two action heads: a lightweight mobility action head $\mathcal{H}_l$ and a mobile manipulation action head $\mathcal{H}$. The lightweight mobility action head $\mathcal{H}_l$—comprising $N$ stacked DiT blocks followed by a final MLP with a total of only 170 million parameters—takes the features from all three modalities—images, point clouds, and language as its input. It aims to extract latent mobility feature that captures the representation of mobile base control, which then serves as mobility prior for the mobile manipulation action head $\mathcal{H}$ (refer to Section 3.3). Beyond the latent mobility feature, the mobile manipulation action head $\mathcal{H}$ also takes the reweighted multimodal features—adaptively adjusted according to the varying perceptual demands of different mobile manipulation states—as input (refer to Section 3.4), and outputs both mobile base and manipulator actions to control the movement of robot. The detailed model parameter summary is shown in Appendix G.

### 3.3 Mobility-to-Body Conditioning

The core premise of this mechanism is that mobile base motion provides essential contextual priors for effective manipulator planning [9]. To explicitly capture the substantial influence of base movement on manipulator control and reduce error accumulation in high-degree-of-freedom (DoF) systems, we propose the Mobility-to-Body Conditioning mechanism. Its goal is to incorporate prior knowledge of base motion into full-body action prediction, facilitating robust and coordinated control between the mobile base and the manipulator.

Specifically, before training the full AC-DiT model $\pi$, we first pretrain all modality encoders together with the lightweight mobility action head $\mathcal{H}_l$ (i.e., the modules highlighted with a gray background in Figure 2) for a few epochs. This pretraining phase aims to enable the model to learn to predict mobile base actions effectively. After pretraining, we enable the lightweight action head $\mathcal{H}_l$ to extract latent mobility features that effectively capture the base motion representation. These features are then used as informative mobility-related priors for the subsequent full-body action prediction.

Next, we optimize the full AC-DiT model, which employs the mobile manipulation action head $\mathcal{H}$ to predict both mobile base and manipulator actions. These predictions are conditioned not only on features extracted from all modalities but also on the **latent mobility features** generated by the pretrained lightweight mobility action head $\mathcal{H}_l$. Specifically, to extract these latent mobility features, we collect the action tokens output by the final DiT block in the lightweight mobility action head $\mathcal{H}_l$ (prior to the final MLP layer) at each of the five denoising time steps. These tokens are then concatenated to form the latent mobility feature $\mathbf{F}_m$, consisting of mobile base action representation. This latent feature is concatenated with the multimodal features, and the combined feature is fed into the cross-attention layers of mobile manipulation action head $\mathcal{H}$, enabling the generation of both mobile base and manipulator actions conditioned on the mobile base prior. By doing so, this conditioning mechanism enables the manipulator to plan its actions with awareness of base movement, thereby enhancing coordination and improving the overall effectiveness of whole-body control.

### 3.4 Perception-Aware Multimodal Adaptation

To address the diverse perceptual needs across different stages of mobile manipulation, such as relying on semantic, rich 2D information for localization and geometrically precise 3D information for interaction—we design a perception-aware multimodal adaptation mechanism. This mechanism dynamically computes the similarity of 2D images and 3D point cloud modalities relative to language instructions, then adaptively assigns different importance weights to generate stage-specific visual representations.

Specifically, we firstly use three MLPs projectors ($\text{Proj}_{2D}, \text{Proj}_{3D}$ and $\text{Proj}_\ell$) to project 2D visual feature $\mathbf{F}_{2D}$ consisting of three camera views, 3D geometric features $\mathbf{F}_{3D}$, and language feature $\mathbf{F}_\ell$ into a shared latent space. We then compute the cosine similarity between each visual modality feature—specifically, the 2D image features from three views (front, left, and right) and the 3D point cloud feature—and the language feature. These similarity scores are subsequently normalized to obtain the importance weights for each modality at the current stage, denoted as $w = w_i^f, w_i^l, w_i^r, w_i^p$. The formulation is as follows:

$$\{w_i^f, w_i^l, w_i^r\} = \cos(\text{Proj}_{2D}(\mathbf{F}_{2D}), \text{Proj}_\ell(\mathbf{F}_\ell)) \tag{1}$$

$$w_i^p = \cos(\text{Proj}_{3D}(\mathbf{F}_{3D}), \text{Proj}_\ell(\mathbf{F}_\ell)) \tag{2}$$

$$F_w^f = w_i^f F_{2D}^f, \quad F_w^l = w_i^l F_{2D}^l, \quad F_w^r = w_i^r F_{2D}^r, \quad F_w^p = w_i^p F_{3D}^p \tag{3}$$

$$F_v = \text{Concat}(F_w^f, F_w^l, F_w^r, F_w^p) \tag{4}$$

In this formulation, $w_i^*$ is a scalar weight and $F_{2D/3D}^*$ are feature vectors, resulting in scalar-vector multiplication. The reweighted visual features are then concatenated to form the perception-aware visual representation $F_v$. This weighted visual feature $F_v$, together with language features $F_l$ and latent mobility features $F_m$, serves as the conditioning input for the mobile manipulation action head. This aims to measure the relevance of different visual observations to the different mobile manipulation stages. For example, it can adaptively assign less weight to left-wrist or right-wrist view images when it shows limited information (e.g., capturing floor or wall), or taking more 3D geometry feature into account when it is about to manipulate the target object. We then apply importance weights on corresponding features to obtain perception-aware visual features $\mathbf{F}_v$, together with language feature $\mathbf{F}_\ell$ and extracted latent mobility feature $\mathbf{F}_m$, forming the conditioning input for the mobile manipulation action head to predict both mobile base and manipulator actions. In this way, different importance weights are adaptively assigned to each visual input, satisfying the distinct perceptual demands at various stages of the mobile manipulation process and enabling AC-DiT to leverage appropriate perceptual priors for robust action prediction.

### 3.5  Training Objectives

During pretraining lightweight mobility action head to extract latent mobility feature, we only finetune the LoRA adapter on 3D encoder and the mobility action head, while keeping the rest of the model frozen. The predicted mobile base actions are supervised under denoising MSE loss. When training the model to predict both mobile base and manipulator actions, we update modules with a fire icon in Figure 2, except those from the pretrained SigLIP model. The predicted mobile base and manipulator actions are also supervised by denoising MSE loss.

## 4  Experiments

In Section 4.1, we compare the mobile and bimanual manipulation capabilities of our method against prior approaches in simulation environments. Section 4.2 presents both quantitative and qualitative results of AC-DiT in real-world scenarios. Finally, Section 4.3 evaluates the effectiveness of each component through detailed ablation studies.

### 4.1  Simulation Experiments

#### 4.1.1  Mobile Manipulation Experiments

**Benchmark**. We employ the Maniskill-Hab (MSHab) [12] simulation benchmark, built upon the Sapien simulator [52]. MSHab further benefits from the integration of ReplicaCAD's [53] photorealistic scene assets and YCB's [54] standardized object models, enabling rich, interactive environments for diverse manipulation tasks. Our experiments include 7 tasks within the **Set-Table** scenario: *pick_apple*, *place_apple*, *pick_bowl*, *place_bowl*, *open_fridge*, *open_kitchen_counter* and *close_kitchen_counter*. **Data collection**. Following MSHab [12], training trajectories are collected with agents trained by reinforcement learning (RL) algorithms. Specifically, PPO [22] is used for *open* and *close* tasks, while SAC [21] is used for *pick* and *place* tasks. Additional

Table 1: Simulation results on mobile manipulation Maniskill-Hab Benchmark. *Mobile* refers to the target location the agent should navigate to (e.g., Fridge or Counter), while *Manipulation* specifies the task the agent should perform upon arrival (e.g., Pick Apple).

| Mobile | Fridge | Table | Fridge | Counter | Table | Counter | Counter | |
|---|---|---|---|---|---|---|---|---|
| Manipulation | Pick | Place | Open | Pick | Place | Open | Close | |
| Obj.
Alg. | Apple | Apple | Door | Bowl | Bowl | Drawer | Drawer | Mean S.R. |
| ACT | $28.0 \pm 2.2$ | $8.7 \pm 3.3$ | $2.0 \pm 2.2$ | $28.0 \pm 2.4$ | $13.0 \pm 0.8$ | $0.0 \pm 0.0$ | $85.7 \pm 1.2$ | 23.6 |
| DP | $21.3 \pm 3.3$ | $28.0 \pm 8.0$ | $7.3 \pm 5.8$ | $20.7 \pm 3.3$ | $\mathbf{69.3 \pm 3.3}$ | $0.0 \pm 0.0$ | $55.0 \pm 5.7$ | 28.8 |
| 3DP | $0.0 \pm 0.0$ | $31.0 \pm 0.8$ | $0.0 \pm 0.0$ | $20.0 \pm 2.4$ | $32.0 \pm 0.8$ | $0.0 \pm 0.0$ | $68.0 \pm 0.0$ | 21.6 |
| RDT | $12.0 \pm 11.3$ | $32.0 \pm 5.7$ | $82.7 \pm 10.5$ | $10.7 \pm 6.8$ | $18.7 \pm 5.0$ | $44.0 \pm 8.6$ | $\mathbf{100.0 \pm 0.0}$ | 42.9 |
| EquiBot | $5.3 \pm 0.6$ | $11.7 \pm 2.1$ | $17.3 \pm 2.1$ | $16.3 \pm 3.2$ | $4.3 \pm 4.0$ | $3.0 \pm 1.0$ | $53.7 \pm 3.2$ | 16.0 |
| $\pi_0$ | $13.0 \pm 1.6$ | $23.3 \pm 1.7$ | $31.3 \pm 2.6$ | $15.7 \pm 1.2$ | $21.3 \pm 2.6$ | $60.0 \pm 0.8$ | $70.0 \pm 2.2$ | 33.5 |
| AC-DiT | $\mathbf{33.3 \pm 1.9}$ | $\mathbf{33.3 \pm 9.4}$ | $\mathbf{90.7 \pm 5.0}$ | $\mathbf{36.0 \pm 6.5}$ | $17.3 \pm 6.8$ | $\mathbf{81.3 \pm 6.8}$ | $97.3 \pm 1.9$ | $\mathbf{55.6}$ |

details of simulated mobile manipulation experiments are shown in Appendix C.1. We generate 1000 successful demonstrations for each task and ensuring the same demonstrations are used by all baseline algorithms. **Baselines**. Due to the limited availability of open-source implementations for many state-of-the-art baselines [2, 8, 10] for mobile manipulation, which inherently complicates reproducibility, we focus our comparative analysis on end-to-end imitation learning methods for mobile manipulation that are publicly accessible and technically reproducible. Therefore, we adopt three representative methods as our baselines: *Diffusion Policy* [13] (DP), a representative 2D imitation learning approach; *3D Diffusion Policy* [14] (3DP), a representative 3D imitation learning approach; *Robotics Diffusion Transformer (RDT)* [11], a diffusion-based foundation model for generalizable manipulation; *EquiBot* [55], a diffusion-based method for mobile manipulation; and $\pi_0$ [15], a flow-based vision-language-action model capable of mobile manipulation. **Evaluation metric**. For all methods, we evaluate 100 episodes 3 times and then compute the average success rate. We report the mean and standard deviation of success rates across the three runs. A successful test episode is defined as one in which the agent completes the task within a maximum of 200 steps. **Training Details**. To ensure fair comparison across methods, we employ identical observation and action spaces for all algorithms. For the observation space, we adopt a historical sliding window of length 2, incorporating the base velocity and absolute positions of upper-body joints. In addition to align with the recommended settings provided by MSHab and ensure fair comparison, we also incorporate several values into the robot states. These include the goal pose, object pose, end-effector pose relative to the base, and a binary indicator of whether the object is grasped. The action space mirrors this structure but uses a horizon of length 2 and replaces absolute joint positions with relative positional changes, while maintaining absolute base velocities. For baselines, we follow their original settings.

**Result Analysis**. We visualize the executed tasks in the top part of Figure 3. From the results in Table 1, AC-DiT consistently achieves the highest success rates in most tasks, with a mean success rate of 48.7%, outperforming all baselines by a significant margin. For example, AC-DiT excels in challenging tasks such as "go to fridge, then open the door" (90.7%) and "go to counter, then close the drawer" (97.3%), demonstrating its strong capability in precise and coordinated of mobile manipulation. In contrast, methods like DP and 3DP exhibit relatively low performance across most tasks. In particular, 3DP struggles with processing large-scale scene-level point clouds during navigation, which makes it difficult to accurately approach the target region and interact with the target object. While the Robotics Diffusion Transformer (RDT) achieves reasonable success on some tasks, it falls short in others due to its lack of explicit modeling of the coordination between the mobile base and the manipulator. As a result, errors in the mobile base's movement tend to accumulate and adversely affect the manipulator's performance during execution. Since EquiBot is a diffusion algorithm that relies on object-centric point clouds, while our experimental environment supports scene-level point clouds, such point clouds may cause some interference to the algorithm, which could be the reason for its lower scores. As for $\pi_0$ [15], while it demonstrates competitive performance exceeding ACT [6], DP [13], and DP3 [14], it is outperformed by AC-DiT in mobile manipulation tasks. One key reason is that pi0 [15] lacks explicit modeling of the locomotion-manipulation interplay, despite successfully demonstrating the feasibility of end-to-end control in specific scenarios (such as clothing pickup and folding). This performance gap underscores the critical contributions of AC-DiT's perception-aware multimodal adaption mechanism and mobility-to-body conditioning framework, which enable seamless integration of mobility and manipulation through dynamic coordination of sensory-motor loops. Overall, the results indicate that AC-DiT's adaptive

Table 2: Simulation results on dual-arm RoboTwin Benchmark.

| | Handover | Contrainer_Place | Cup_Place | Bottles_Easy | Bottles_Hard | Pick_Apple | Mean S.R. |
|---|---|---|---|---|---|---|---|
| DP | $0.7 \pm 0.5$ | $33.7 \pm 2.9$ | $56.7 \pm 4.1$ | $81.0 \pm 3.3$ | $49.0 \pm 2.2$ | $6.7 \pm 1.2$ | 37.9 |
| 3DP | $82.0 \pm 4.3$ | $73.3 \pm 5.7$ | $74.7 \pm 5.3$ | $83.7 \pm 10.7$ | $61.0 \pm 4.9$ | $65.3 \pm 5.7$ | 73.3 |
| RDT | $85.3 \pm 18.0$ | $56.0 \pm 3.3$ | $48.0 \pm 6.5$ | $93.3 \pm 3.8$ | $68.0 \pm 9.8$ | $69.3 \pm 8.2$ | 70.0 |
| AC-DiT | $\mathbf{100.0 \pm 0.0}$ | $\mathbf{81.3 \pm 8.2}$ | $\mathbf{86.7 \pm 13.6}$ | $\mathbf{100.0 \pm 0.0}$ | $\mathbf{84.7 \pm 4.7}$ | $\mathbf{88.0 \pm 3.3}$ | $\mathbf{90.1}$ |

coordination and perception mechanisms significantly enhance task success across diverse mobile manipulation scenarios.

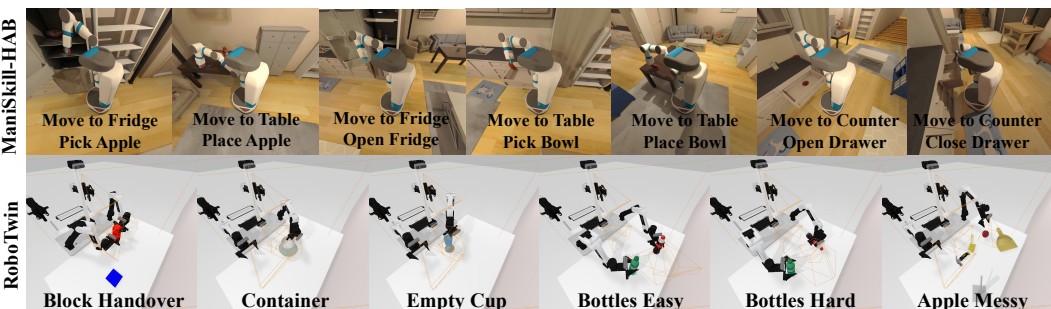

Figure 3: Robot execution visualization of 7 tasks in mobile simulator ManiSkill-HAB and 6 tasks in bimanual simulator RoboTwin.

### 4.1.2 Bimanual Manipulation Experiments

The proposed training mechanisms can be readily extended to dual-arm robotic tasks, enhancing coordination between the arms. Specifically, the perception-aware multimodal adaptation mechanism can be seamlessly applied to bimanual scenarios, while the mobility-to-body conditioning mechanism can be adapted into a dual-arm-to-dual-arm conditioning mechanism. This facilitates cross-arm prior conditioning, enabling each arm to anticipate the other's actions before action prediction, thereby improving bimanual coordination. Therefore, we evaluate the effectiveness of these training mechanisms in the bimanual setting and assess their effectiveness on dual-arm coordination.

**Benchmark**. We use RoboTwin [56] as our simulation environment, which is also built upon the Sapien simulator [52]. We adopt six commonly used dual-arm tasks from RobotWin: *block_handover*, *container_place*, *dual_bottles_pick_easy*, *dual_bottles_pick_hard*, *empty_cup_place* and *pick_apple_messy*. Additional details of simulated bimanual manipulation experiments are shown in Appendix C.2. **Data collection**. In RoboTwin, demonstrations are automatically generated by motion planning tools on the basis of key poses annotated by GPT-4 [16]. We generate 1000 trajectories for each task. **Result Analysis.** We visualize the executed tasks in the bottom part of Figure 3. We adopt the same baselines and evaluation metrics as described in Section 4.1.1, in which 3DP is used by RoboTwin to evaluate bimanual performance, while RDT also supports bimanual manipulation. It can be observed in Table 2, that the Diffusion Policy (DP) performs poorly across all tasks, with an average success rate of only 38.2%. In contrast, though the 3D Diffusion Policy (3DP) and Robotics Diffusion Transformer (RDT) show more stable and consistent performance, our proposed method, AC-DiT, achieves the best results across all tasks, demonstrating its strong bimanual coordination performance. These results demonstrate that our proposed method can be effectively extended to bimanual tasks, enhancing coordination between multiple robotic arms and highlighting the scalability of AC-DiT.

## 4.2 Real-World experiments

**Dataset collection**. In our real-world robot experiments, we employ the Agilex Cobot Magic platform, which consists of four Agilex Piper arms and a Tracer mobile base. In addition to the three Orbbec Dabai cameras located at the wrists and chest positions, we apply an extra RealSense L515 depth camera on the head to capture point clouds. We conduct four mobile manipulation tasks: *mobile_pick_place_bread*, *mobile_pick_place_cumcuber*, *mobile_clean_table*, and *mobile_hang_towel*. These tasks involve various objects and manipulation actions, and they are all long-horizon tasks,

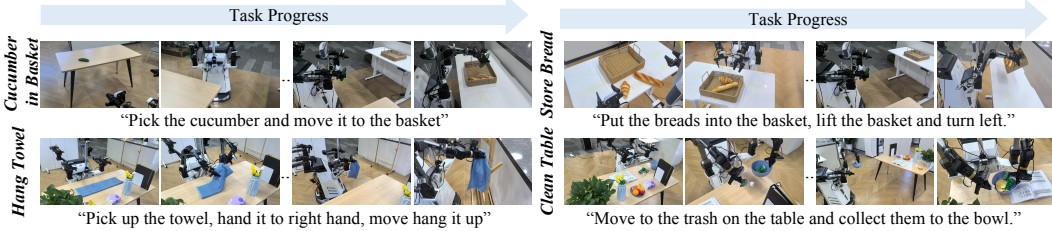

Figure 4: Robot execution progress of AC-DiT in four real-world tasks.

Table 3: Comparison with baselines in the real world. We report the success rate of the overall task (e.g., cucumber in basket) as well as its corresponding sub-tasks (e.g., move, grasp, move, place).

| Task | Cucumber in Basket | | | | Store Breads | | | | | Hang Towel | | | | Clean Table | | | | | |
|---|---|---|---|---|---|---|---|---|---|---|---|---|---|---|---|---|---|---|---|
| Sub-task | Move | Grasp | Move | Place | Grasp | Place | Lift | Move | Place | Grasp | Handover | Move | Hang | Bowl | Bottle | Sprite | Move | Peel | Napkin |
| ACT | 0.0 | | | | 0.0 | | | | | 0.0 | | | | 6.3 | | | | | |
| | 68.8 | 6.3 | 0.0 | 50.0 | 12.5 | 12.5 | 0.0 | 0.0 | 37.5 | **81.3** | 6.3 | 0.0 | 6.3 | 68.8 | 25.0 | 18.8 | 12.5 | 18.8 | 37.5 |
| $\pi_0$ | 31.3 | | | | 25.0 | | | | | 18.8 | | | | 12.5 | | | | | |
| | 75.0 | 53.0 | 62.5 | 68.8 | 68.8 | 56.3 | 81.3 | 75.0 | 68.8 | **81.3** | 56.3 | **68.8** | 43.8 | 81.3 | 56.3 | 75.0 | **68.8** | 62.5 | 50.0 |
| AC-DiT | **50.0** | | | | **56.3** | | | | | **43.8** | | | | **43.8** | | | | | |
| | **93.8** | **75.0** | **68.8** | **93.8** | **87.5** | **100.0** | **81.3** | **87.5** | **81.3** | 75.0 | **81.3** | **68.8** | **68.8** | **100.0** | **75.0** | **81.3** | 56.3 | **81.3** | **75.0** |

including at least 4 subtasks. For each task, we collect 100 spatially generalized demonstration trajectories at a frequency of 30 fps. Due to space limitations, details of the hardware setup, real-world experimental settings, and corresponding results are provided in Appendix A and B, respectively. **Training and Evaluation Details**. We train a single model for all the tasks. For evaluation, we evaluate the model for 17 trials with diverse initial object poses. Given that our real-robot tasks are all long-horizon in nature, we conduct tests on each atomic subtask of these long-horizon tasks to enable a more granular comparison of each method. Note that if the previous sub-task fails, we manually reset it to allow evaluation of its performance in the next sub-task. However, for the overall task success rate, all sub-tasks must be completed successfully.

**Quantitative Results**. As shown in Table 3, we compare AC-DiT with ACT [6], a method implemented for bimanual mobile manipulation, and $\pi_0$ [15], a vision-language-action model also applied to bimanual mobile manipulation. Same as the results in simulators, AC-DiT performs the best across all tasks, showing its capability of mobile bimanual manipulation in real-world. **Qualitative Results**. As shown in Figure 4, we visualize the execution of our real-world tasks, all of which are long-horizon and involve multiple steps. These tasks are designed to showcase the capability of the proposed AC-DiT in handling complex, multi-stage mobile manipulation. Especially in the bimanual task of hanging towel, the dual robot arms must first cooperate to hand over the towel, while the mobile base simultaneously navigates to the hanger. During this process, coordination between the mobile base and the manipulators is essential to ensure the towel is stably transferred to the hanger. As the agent approaches the hanger, it must then accurately hang the towel in place. Additional qualitative results are shown in Appendix E and the failure cases are analyzed in Appendix F.

As shown in Figure 5, we further visualize how the importance weights of different visual inputs change during the real-world clean table task to show the effectiveness of perception-aware multimodal adaptation mechanism, as shown in Figure 5. When the robot approaches the table and prepares to grasp the crumpled tissue, the importance weights of the right-wrist camera and the point cloud increase. This is because these views provide rich semantic and spatial information about the target object, which helps the model accurately predict grasping actions. In contrast, the

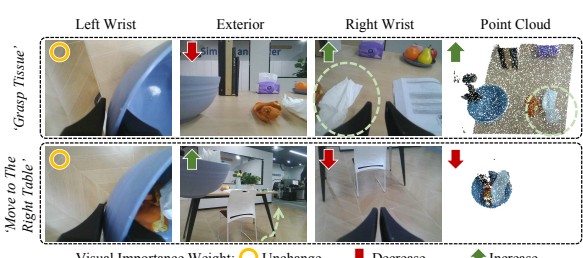

Figure 5: Effectiveness of Perception-aware Multimodal Adaption mechanism.

importance weight of the external view decreases, as the tissue is not clearly visible from that angle. The left-wrist view's importance remains unchanged, since the left arm consistently holds a bowl and does not need to take action during this phase. While the robot is moving, the external camera's importance increases, as it provides a wide field of view that supports navigation and spatial awareness. Meanwhile, the right-wrist camera's weight decreases because its highly localized perspective offers limited help for movement.

## 4.3 Ablation Study

We use MSHab as the benchmark for our ablation study. Different from the main experiments, which were conducted with three independent runs for each configuration and reported the mean success rate and standard deviation, we only report the results from a single experimental run in this section due to computational limitations. Table 4 presents the results of our ablation study evaluating the impact of different components in the

Table 4: **Ablation study**. 2D and 3D represent whether take images and points cloud as input respectively. MBC and MA denote the proposed **M**obility-to-**B**ody **C**onditioning (MBC) and **P**erception-aware **M**ultimodal **A**daption (PMA) mechanisms, respectively.

|          | 2D | 3D | PMA | MBC | Mean | Gain |
|----------|----|----|-----|-----|------|------|
| Exp1     | ✓  | -  | -   | -   | 37.5 | -    |
| Exp2     | ✓  | ✓  | -   | -   | 44.8 | +7.3 |
| Exp3     | ✓  | ✓  | -   | ✓   | 47.0 | +9.5 |
| Exp4(ours) | ✓ | ✓ | ✓   | ✓   | 49.0 | +11.5 |

proposed AC-DiT framework. In Exp1, using only 2D visual inputs results in a baseline mean success rate of 37.5%. Introducing both 2D and 3D modalities in Exp2 yields a notable improvement (44.8%), demonstrating the benefit of multimodal fusion. Exp3 further incorporates the Mobility-to-Body Conditioning (MBC) mechanism, boosting the performance to 47.0% and suggesting that explicitly modeling the influence of the mobile base improves coordination. Finally, Exp4 integrates both MBC and the Perception-Aware Multimodal Adaptation (PMA) mechanism, achieving the best performance (49.0%). This confirms that adaptively weighting multimodal features further enhances the system's ability to handle complex, long-horizon tasks by taking into account different perception requirements at different mobile manipulation stages. Overall, each component contributes positively, with the full model showing a total gain of 11.5 In Appendix D, we explore the impact of how the conditions are injected into the DiT, and examine the impact of how to establish the conditioning relationship.

## 5 Conclusion and Limitations

We present AC-DiT (Adaptive Coordination Diffusion Transformer), an end-to-end framework that advances mobile manipulation by explicitly modeling the coordination between the mobile base and the manipulator, while dynamically adapting to stage-specific perception needs. To this end, we introduce two key mechanisms: a mobility-to-body conditioning module, which leverages latent mobility representations to improve coordination and reduce error accumulation; and a perception-aware multimodal adaptation module, which assigns adaptive weights to 2D and 3D visual inputs based on stage demands. Extensive experiments in both simulation and real-world settings demonstrate that AC-DiT consistently outperforms state-of-the-art baselines, showcasing its effectiveness in robust mobile manipulation. As for limitations, since AC-DiT adopts imitation learning, the performance would fluctuate with data quantity and quality, as well as inheriting suboptimal or biased demonstrations. Finally, we state the social impact of our work in Appendix H.

# 6 Acknowledgements

This work was supported by the National Natural Science Foundation of China (62476011).

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

**Appendix A Robot Hardware Details.** In this section, we provide a detailed description of the robotic platform used in our experiments, including the Agilex Piper robotic arms, the Tracer mobile base, and the camera configurations. Specific joint ranges, control modes, and sensor parameters are outlined.

**Appendix B Real-World Experiment Details.** In this section, we provide comprehensive details on the real-world experiments. It covers the robot Configuration and Assets (Appendix B.1); detailed Task Descriptions (Appendix B.2) for the four complex manipulation tasks; specifics on Computational Resources and Time Cost (Appendix B.3), such as GPU configuration, training stages, iteration counts, model parameters, and memory usage; and the achieved Control Frequency (Appendix B.4).

**Appendix C Simulation Experiment Details.** In this section, we provide qualitative and quantitative results of AC-DiT in simulated tasks. We highlight key challenges in simulation and further demonstrate the effectiveness of our proposed Mobility-to-Body Conditioning mechanism and Perception-Aware Multimodal Adaptation mechanism.

**Appendix D Additional Ablation Study.** In this section, we present an in-depth ablation study investigating the impact of how the conditions are injected into the DiT and the impact of how to establish the conditioning relationship.

**Appendix E Additional Qualitative Results.** To further validate the effectiveness of the Perception-aware Multimodal Adaptation mechanism, we supplement the visualization experiment from Figure 5 in the main paper with additional results.

**Appendix F Failure Case Analysis.** In this section, we discuss common failure modes observed in real-world experiments, categorizing them into bimanual coordination errors, locomotion control failures, and issues caused by poor manipulation triggering.

**Appendix G Model Parameter Breakdown.** In this section, we provide a comprehensive breakdown of the model parameters, explaining the rationale behind selecting the Lightweight Mobility Head for base control and the Large Mobile Manipulation Head for whole-body control based on task complexity and action space dimension.

**Appendix H Broader Impact.** In this section, we reflect on the potential broader impact of our work, highlighting its contribution to the development of end-to-end mobile manipulation systems and its societal implications.

# A   Robot Hardware Details

Table 5: Agilex Piper Arm Position Ranges

| Joint Name | Range |
|---|---|
| J1 | $-154° \sim 154°$ |
| J2 | $0° \sim 195°$ |
| J3 | $-175° \sim 0°$ |
| J4 | $-106° \sim 106°$ |
| J5 | $-75° \sim 75°$ |
| J6 | $-100° \sim 100°$ |

Table 6: Configurations for cameras

| Parameter | Value |
|---|---|
| Orbbec Dabai Cameras | |
| FOV (H×W) | $80° \times 66°$ |
| Frequency | 120 fps |
| Realsense L515 Camera | |
| FOV (H×W) | $70° \times 43°$ |
| Frequency | 30 fps |

**Arms**. As illustrated in Figure 6a, the Cobot Magic platform incorporates four 6-DoF Agilex Piper robotic arms. The two puppet arms for inference and execution each feature a parallel gripper with a 110 mm stroke. The robotic arms operate in joint position control mode, and the motion range of each joint is detailed in Table 5.

**Mobile base**. The mobile base, named Tracer, is wheel-based and capable of in-situ steering and forward/backward locomotion. It is equipped with two active drive wheels and four passive idler wheels. The Tracer is controlled via a 2-DOF velocity controller, governing two independent parameters: forward velocity and rotational velocity. The base achieves a maximum speed of 1.8 meters per second.

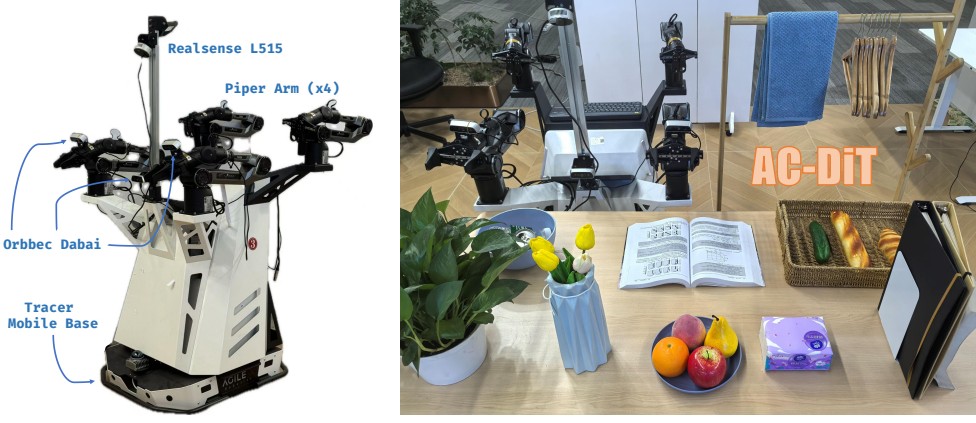

| (a) Robot hardware configuration. | (b) Real-World assets. |

Figure 6: Robot hardware configuration and real-world assets.

**Cameras**. As shown in Figure 6a, the Cobot Magic comes pre-installed with three Orbbec Dabai cameras. To capture more precise point cloud data, we add an additional Intel RealSense L515 depth camera at the head. Notably, the two central cameras have specific design requirements for their positions and orientations. The central Dabai camera is mounted at chest level with a frontal view to enable the robot to observe a sufficiently wide field of view during mobility. Meanwhile, the L515 camera is positioned at the head with an overhead perspective to fully observe the workspace during the manipulation phase. The specific parameters of the two cameras are shown in Table 6.

## B    Real-World Experiment Details

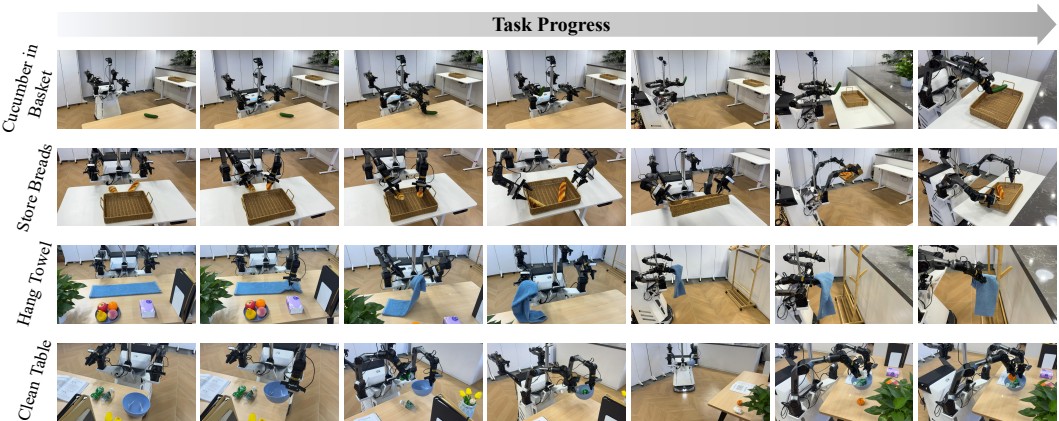

Figure 7: Robot execution progress of AC-DiT in real-world tasks.

### B.1    Configuration and Assets

The robot hardware configuration and real-world assets are shown in Figure 6a and Figure 6b, respectively. Except that AC-DiT uses the RGB information from three Dabai cameras and the depth information from an L515 camera, all other baseline algorithms use the RGB information from all four cameras. Besides, as shown in Figure 7, we provide a more detailed visualization of the progress of real-world tasks.

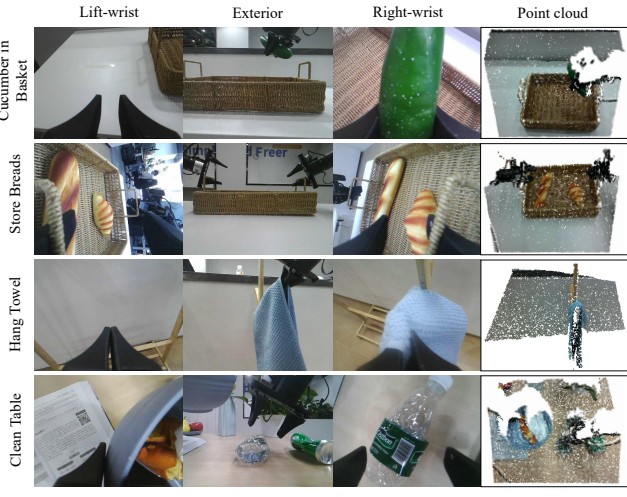

Figure 8: Observation examples in real-world tasks.

## B.2 Task Descriptions

Detailed descriptions of all real-world robot tasks are listed as follows:

1. *Cucumber in basket.* The robotic setup includes a primary worktable in front of the robot holding a cucumber and a secondary table on the robot's left side with a basket for object placement, where the robot starts at an initial distance from the primary worktable with its arm in the rest pose. The robot must first navigate to the cucumber on the primary table, execute a grasp action to acquire it, then retreat slightly, perform a left turn maneuver to reorient toward the secondary table, navigate to the basket, and precisely deposit the cucumber into it before retracting its arm to the predefined rest position to complete the task.

2. *Store bread.* The robotic workspace comprises a primary worktable positioned in front of the robot, equipped with two bread items and a basket, and a secondary left-side table initially empty. The robot starts at an initial position relative to the primary worktable with its arms in the rest configuration. The task requires the robot to first navigate to the primary worktable and execute bilateral arm manipulation to transfer the bread items into the basket, followed by lifting the basket with coordinated dual-arm motion. Subsequently, the robot performs a backward translation, a left rotational reorientation, and navigates to the secondary table, where it deposits the basket with precise placement to complete the task.

3. *Hang towel.* The robotic task environment consists of a workdbench positioned in front of the robot, bearing a towel as the target object, and a hanging rack located on the robot's left lateral side. The robot initiates the task at an initial pose in front of the workbench with its arms in the rest configuration. The operational sequence begins with the robot using its left arm to execute a precision grasp of the towel, followed by a bimanual transfer to the right arm. The robot then performs a minor backward translation, a left rotational reorientation, and navigates to the hanging rack, where it executes a towel-hanging action with precise placement on the rack. The task is concluded by the robot returning both arms to their predefined rest positions.

4. *Clean table.* The robotic workspace features a workbench equipped with four waste objects: a Sprite bottle, a Cestbon bottle, orange peel, and a crumpled tissue. A large bowl and two randomly positioned waste objects are initially placed on the workbench's left side, with the remaining two objects on the right side. At task initiation, the robot is positioned on the left side of the workbench with its arms in the default configuration. The task sequence unfolds as follows: first, the robot uses its left arm grasp to lift the large bowl, then utilizes its right arm to transfer the two left-side waste objects into the bowl. Subsequently, the robot performs a short backward translation, reorients toward the front-right direction, and navigates to the workbench's right side, where it continues using its right arm to deposit the remaining two objects into the bowl, thereby completing the task.

### B.3 Computational Resource and Time Cost

Our proposed approach was trained on a server equipped with eight NVIDIA A100 (80 GB) GPUs. The training process consisted of two stages: an initial Mobility pretraining phase (Stage 1) comprising 20,000 iterations, followed by training of the full AC-DiT model (Stage 2), which comprised 30,000 iterations. The lightweight mobility head of our model contains 170 million parameters, while the full AC-DiT model has a total of 1.2 billion parameters. During training with a batch size of 16, the system utilized approximately 60 GB of GPU memory. Inference requires less memory, consuming about 20 GB of GPU memory.

### B.4 Control Frequency

The model achieves an inference speed of 5Hz on a single NVIDIA RTX 4090 GPU. This speed, combined with action chunking, allows for a robot control frequency of 30Hz. The total processing time for a single control step is 0.2s, which is composed of the model inference time (0.16s), perception information acquisition (0.03s), and point cloud computation (0.01s).

## C    Simulation Experiment Details

### C.1    ManiSkill-Hab Simulationi Environment

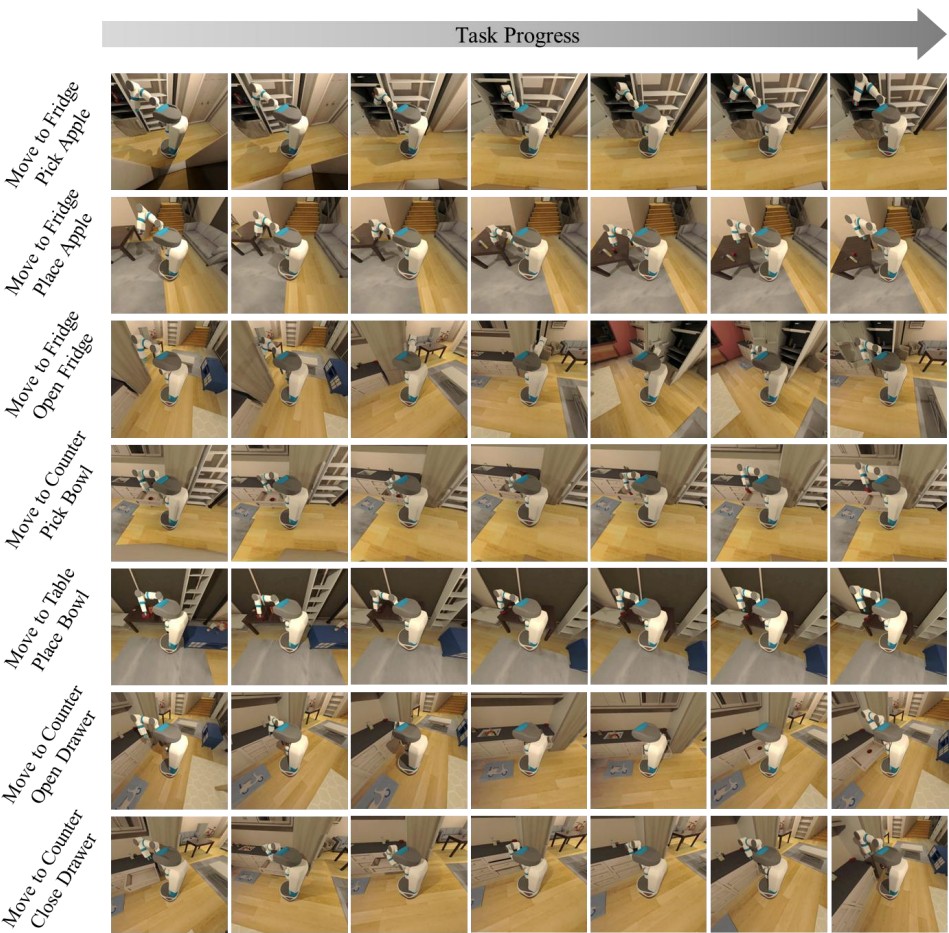

Figure 9: Robot execution progress of AC-DiT in seven MSHab simulation tasks.

Figure 9 illustrates the process of AC-DiT executing simulation tasks in MSHab environments and Figure 10a shows the observation examples of each task in MSHab. Notably, in the MSHab simulator, since trajectories are collected by the RL agent, the mobile base and manipulator frequently exhibit

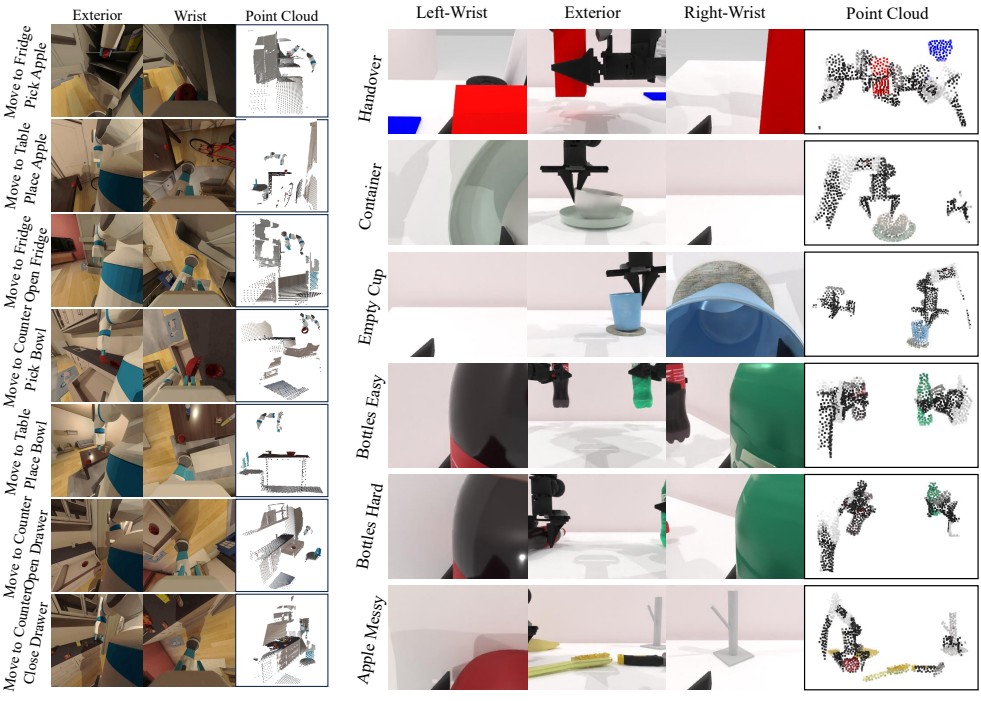

(a) Observation examples in MSHab simulation tasks.

(b) Observation examples in RoboTwin simulation tasks.

Figure 10: Observation examples in MSHab and RoboTwin simulations.

concurrent motion. This scenario rigorously evaluates the coordination capability between the mobile base and manipulator while further underscoring the efficacy of the mobility-to-body condition mechanism. Specifically, in the task depicted in Figure 9, the mobile base and the manipulator perform concurrent motions to approach the target object, yet the gripper still accurately executes the manipulation at the optimal timing. This demonstrates that AC-DiT's modeling of coordination plays a critical role in achieving such seamless integration of locomotion and manipulation.

## C.2 RoboTwin Simulation Environment

Figure 11 illustrates the process of AC-DiT executing simulation tasks in RoboTwin environments and Figure 10b present the observation examples of each task in RoboTwin. Notably, the *Handover* task necessitates coordinated collaboration between two arms rather than independent actions, underscoring the critical requirement for bimanual coordination. This further highlights that AC-DiT's dual-arm-to-dual-arm conditioning mechanism in RobotWin—derived from the mobility-to-body conditioning mechanism—plays a pivotal role by providing each arm with short-term motion priors, thereby enhancing bimanual coordination.

## D  Additional Ablation Study

Table 7: Ablation Study. We explore the impact of how the conditions are injected into the DiT.

| Condition Injection | Mean |
|---|---|
| Cross-Attention Alternatively | 43.7 |
| Cross-Attention | 44.8 |

Table 8: Ablation Study. We explore the impact of how to establish the conditioning relationship.

| Condition Mechanism | Mean |
|---|---|
| Up-to-Body | 47.3 |
| Mobility-to-Body | 49.5 |

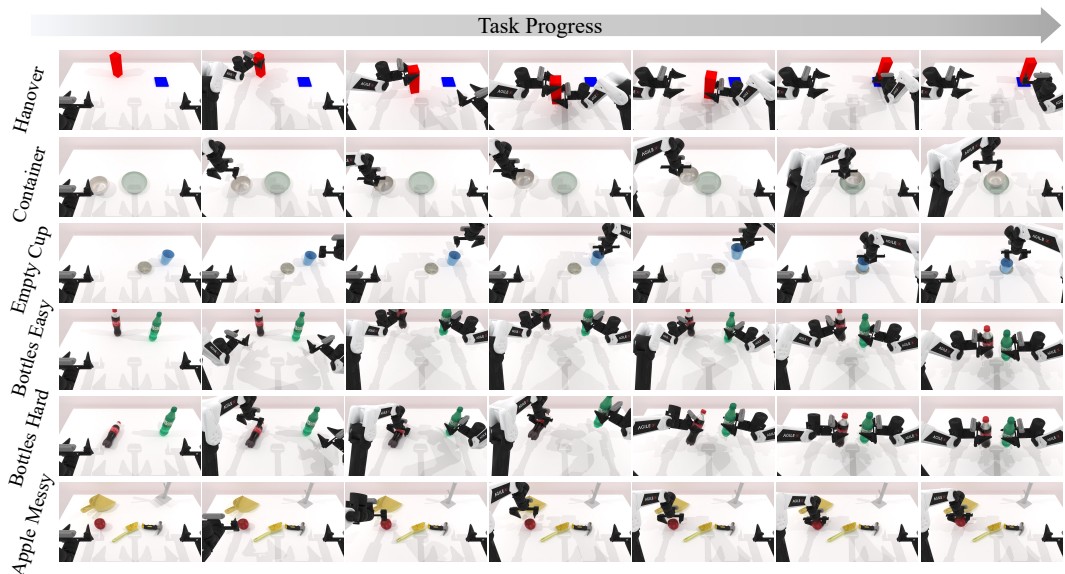

Figure 11: Robot execution progress of AC-DiT in seven RoboTwin simulation tasks.

**The impact of how the conditions are injected into the DiT.** We empirically evaluated two different methodologies for condition injection into DiT. First, following the RDT framework [11], we use cross-attention for alternating condition injection. Second, we implement cross-attention-based condition injection with concatenation of all conditional features in each injection step [57]. As shown in Table 7, cross-attention variants (with or without alternatively injection) exhibit comparable performance, with the non-alternatively-injection scheme achieving marginally superior results due to its higher information throughput relative to the alternatively-injection one.

**The impact of how to establish the conditioning relationship.** In AC-DiT, we first extract a latent mobility feature to serve as a condition for mobile manipulation action prediction, as shown in the "Mobility-to-Body" setting in Table 8. In this section, we compare this approach with using a latent manipulator feature—specifically, the arm joint position—as the condition, corresponding to the "Up-to-Body" setting in Table 8. We observe that using the latent mobility feature outperforms using the latent manipulator feature. This is because the mobile base actions have a significant impact on manipulator control; errors in the mobility prediction can accumulate and propagate to the manipulator stage. Therefore, we adopt the strategy of first extracting the latent mobility feature as the condition to alleviate potential error accumulation and enhance coordination.

# E  Additional Qualitative Results

**Adaptive importance weights analysis.** To further validate the effectiveness of the Perception-aware Multimodal Adaptation mechanism, we supplement the visualization experiment from Figure 5 in the main paper with additional results, as shown in Figure 12. We visualize the observations at three different time steps. As shown, the importance of the left-wrist camera remains unchanged, since the left arm consistently holds a bowl and does not need to take action during this phase. In contrast, at time steps 1 and 3, the importance weights of the right-wrist camera and the point cloud increase obviously, as they provide critical information for completing the manipulation task. Meanwhile, the exterior camera maintains a relatively high importance, as it still captures both the manipulated object and the robot arms from a broader perspective. In contrast, during the movement phase at step 2, the importance weight of the point cloud drops significantly, due to the limited sensing range of the depth camera.

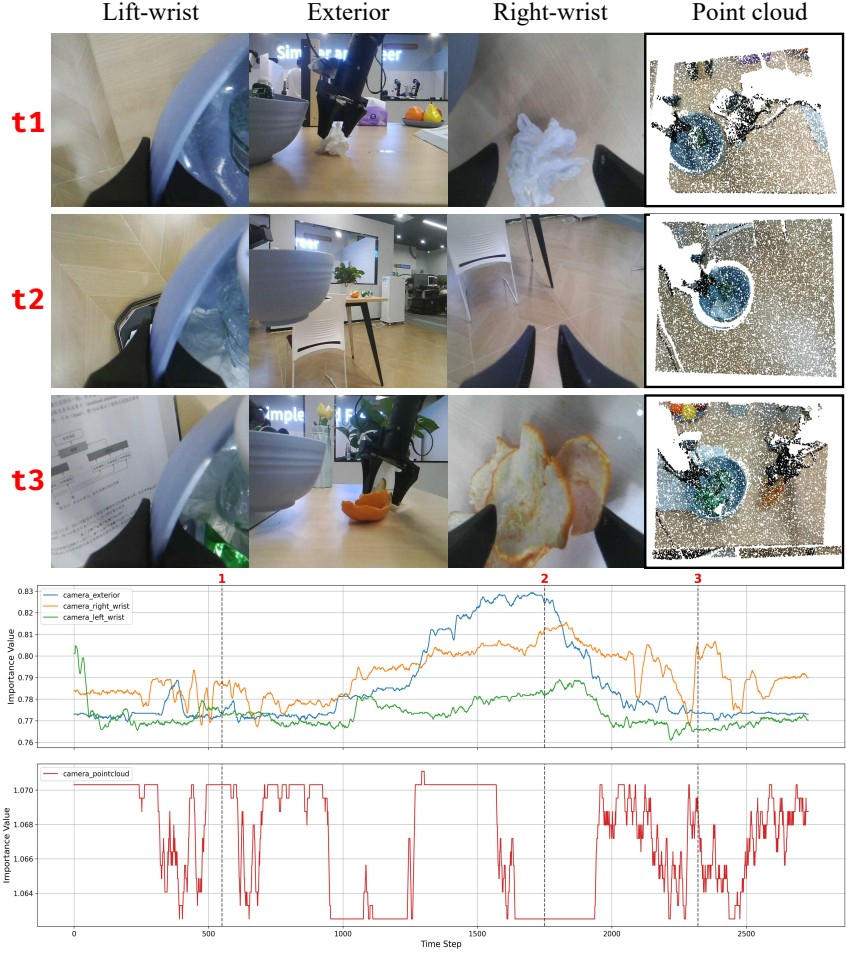

Figure 12: Visualization of Perception-aware Multimodal Adaption mechanism on a whole episode.

## F Failure Case Analysis

Despite extensive physical robot experiments, we identify three primary error categories affecting AC-DiT's performance. First category: **Bimanual coordination errors**. Bilateral arm collaboration is required for tasks such as transferring a towel from the left hand to the right hand, where the right hand occasionally fails to grasp the correct pose, or the right arm inaccurately deposits waste into a bowl, causing objects to fall outside due to minor positional deviations. Second category: **Locomotion errors**. Since robot movement is controlled by a velocity controller, whose actual motion results from the integration of velocity commands over time, cumulative errors may occur. The model sometimes outputs excessively high velocities, leading the robot into unrecoverable error states, or insufficient velocities that prevent chassis movement. Mitigating this issue relies on dataset quality—ideally, data should be collected by a single operator with minimal velocity fluctuations maintained at a suitable constant value during recording. Third category: **Suboptimal manipulation triggering due to poor locomotion termination**. When the robot stops at a state outside the dataset's distribution, the subsequent manipulation primitives often fail, as the end pose does not align with the precondition for successful object interaction.

## G Model Parameter Summary

We display the comprehensive parameter breakdown in Table 9, specifically detailing all components shown in Figure 2 of the main paper. The relative small DiT architecture (Lightweight Mobility Head, 170M DiT) is selected for mobile base control as it has relatively fewer output degrees of freedom

Table 9: Model Parameter Summary.

| Component | Parameters | Architecture Details |
|---|---|---|
| Image Encoder | 428M (frozen) | SigLIP-based multi-view visual feature extraction |
| Text Encoder | 450M (frozen) | SigLIP-based language instruction encoding |
| 3D Encoder w/. LoRA | 430M (0.09% LoRA) | Point cloud processing with LoRA adaptation |
| Perception-aware Multi-modal Adaptation | 1M | Linear Projectors |
| Mobile Manipulation Action Head (DiT) | 1.2B | Main DiT-based head for whole-body action prediction |
| Lightweight Mobility Action Head (DiT) | 170M | DiT blocks for base motion prediction |
| **Total Trainable** | **∼1.37B** | **Total trainable parameters** |
| **Total Model** | **∼2.7B** | **Including frozen foundation models** |

(linear/angular velocities), providing sufficient modeling capacity while reducing computational overhead. The relative large architecture (Large Mobile Manipulation Head,1.2B DiT) is selected for the more complex mobile manipulation task, where higher parameter count provides better robustness and generalization capability needed for coordinated whole-body control with higher-dimensional action spaces.

## H  Broader Impact

Our work develops an end-to-end mobile manipulation framework based on DiT [57, 11] by explicitly modeling the collaborative relationship between the mobile base and the manipulator, and by adaptively adjusting weights according to perceptual needs at various stages of mobile manipulation. This research aims to innovate by extending end-to-end models to mobile manipulation tasks, and it does not pose direct societal impacts. We hope our efforts can foster research in the field of end-to-end control for mobile manipulation while promoting the healthy, controlled, and sustainable development of the entire field.

