# OpenReview forum: "AC-DiT: Adaptive Coordination Diffusion Transformer for Mobile Manipulation"
_NeurIPS.cc/2025/Conference — NeurIPS 2025 poster_

### Official Review · Reviewer_3fe8 · 2025-06-23

**Clarity:** 3
**Significance:** 3
**Originality:** 3
**Rating:** 4
**Confidence:** 4

**Summary:**

This paper proposes AC-DiT, a novel diffusion transformer framework designed for mobile manipulation. The method introduces two key modules: (1) **Mobility-to-Body Conditioning**, which uses latent mobility features to guide whole-body action generation, and (2) **Perception-Aware Multimodal Adaptation**, which dynamically adjusts the importance of 2D and 3D visual modalities based on task phase. The approach is validated through extensive simulation and real-world experiments, showing improved performance over imitation learning baselines like DP, RDT, and π₀. AC-DiT aims to address coordination challenges in mobile manipulation under vision-language guidance.

**Questions:**

* **1) How does AC-DiT compare with mobile manipulation-specific baselines like Mobile-ALOHA, TidyBot++, or EquiBot?**
The current comparisons seem limited to general-purpose or tabletop-oriented methods.

* **2) Are the two proposed modules (Mobility-to-Body Conditioning and Perception-Aware Adaptation) significantly better than trivial attention mechanisms?**
Could similar performance be achieved using standard cross-modal or self-attention?

* **3) Can the same AC-DiT policy generalize across diverse mobile manipulation tasks without task-specific fine-tuning?**
If not, how scalable is the approach for real-world multi-task deployment?

**Ethical Concerns:**

["NO or VERY MINOR ethics concerns only"]

**Final Justification:**

Given the solid empirical results, rigorous ablations, and the additional clarifications provided, I maintain my original positive assessment of the paper and believe the work makes a valuable contribution to the field of mobile manipulation.

**Limitations:**

**Limited Methodological Novelty:**
The core mechanisms are customized attention strategies without strong theoretical justification or architectural innovation.

**Incomplete and Possibly Unfair Comparisons:**
The evaluation lacks direct comparison to mobile manipulation-focused baselines, weakening claims of state-of-the-art performance.

**Paper Formatting Concerns:**

No potential formatting issues.

**Quality:**

3

**Strengths And Weaknesses:**

**Strengths**
* **1) Problem Motivation is Well-Stated:** The paper clearly identifies critical coordination and perceptual gaps in prior mobile manipulation methods and proposes concrete solutions.
* **2) Simple Yet Effective Architecture:** The proposed modules are computationally lightweight and integrate seamlessly into diffusion-based transformers, contributing to scalability.
* **3) Strong Empirical Results:** AC-DiT outperforms multiple baselines across both simulated and real-world tasks, demonstrating its practical value.
* **4) Detailed Ablation Studies:** The experiments carefully evaluate the impact of each component (e.g., 2D/3D fusion, conditioning mechanisms), reinforcing the modular design’s benefits.

**Weaknesses**
* **1) Method Novelty is Limited:** Both proposed modules—Mobility-to-Body Conditioning and Perception-Aware Adaptation—are essentially task-specific adaptations of attention-based conditioning mechanisms. The core architectural contribution is somewhat incremental.
* **2) Unfair Baseline Comparison:** AC-DiT is compared primarily against methods like ACT, DP, and RDT, which were designed for tabletop or bimanual tasks, not mobile manipulation. Missing comparisons to Mobile-ALOHA [1], TidyBot++ [2], or EquiBot [3] weakens the empirical claims.
* **3) Writing and Typographical Errors:** Several typos and formatting issues are present, such as "RoboTwen" (should be RoboTwin, line 270), "issue" spelled as "tissue" (line 322), and wrong reference to “Figure 3” where it should be “Table 3” (line 296), indicating the need for a more careful proofreading.
* **4) Limited Theoretical Rigor:** The paper lacks a formal analysis or theoretical justification for why the proposed conditioning mechanisms are superior to simpler alternatives, such as vanilla self-attention across modalities or joints.

```
[1] Mobile ALOHA: Learning Bimanual Mobile Manipulation with Low-Cost Whole-Body Teleoperation, CoRL 2024
[2] TidyBot++: An Open-Source Holonomic Mobile Manipulator for Robot Learning, CoRL 2024
[3] EquiBot: SIM(3)-Equivariant Diffusion Policy for Generalizable and Data Efficient Learning, CoRL 2024
```

In a word, while the paper demonstrates solid engineering and compelling results, its conceptual contributions may be viewed as modest, and comparisons with stronger, more relevant baselines are missing. Strengthening the theoretical motivation and improving experimental fairness could help to raise its final score.

---

> ### Author Rebuttal · Authors · 2025-07-31
>
> # Response to Reviewer 3fe8
>
> We appreciate the reviewer's constructive feedback and positive assessment of our contributions. We provide detailed responses to the raised questions in the following.
>
> ## **Response to Weakness 1 & Limitation 1: Clarification on Innovation**
>
> **Clarification on Innovation Focus:** Our core contribution is not task-specific adaptations of attention-based architectures, but rather the overall **framework design** tailored for generalistic mobile manipulation. we propose a mobile manipulation paradigm that can be applied across various architectures. We introduce a new formulation for mobile manipulation that addresses critical coordination and perceptual gaps. Our two proposed innovations (MBC and PMA) present a first attempt to explicitly model the unique challenges of mobile manipulation. **While our current work applies these to a diffusion-based transformer, the underlying principles have the potential to benefit various mobile manipulation architectures that utilize multimodal perception.** The key insight is that mobile manipulation requires fundamentally different coordination mechanisms compared to tabletop tasks.
>
> **Novelty of Our Contributions:**
>
> 1. **Mobility-to-Body Conditioning (MBC):** We make the **first attempt** to formulate mobility-to-body control that explicitly models the causal dependency between mobile base motion and manipulator dynamics in end-to-end learning frameworks. Unlike existing approaches that treat base and manipulator actions independently or simply concatenate them, our **novel conditioning paradigm** extracts latent mobility features first and uses them as explicit priors for whole-body action prediction. This represents a **fundamental departure** from conventional joint optimization strategies by establishing a hierarchical conditioning relationship that mirrors the physical causality inherent in mobile manipulation systems.
> 2. **Perception-Aware Multimodal Adaptation (PMA):** Different from other works that treat all modalities equally throughout the manipulation process, we make the **first attempt** to dynamically adapt multimodal fusion based on task-stage semantics. Our **novel insight** is that mobile manipulation exhibits distinct perceptual requirements across different phases—semantic understanding during navigation versus geometric precision during interaction. Rather than using static fusion weights or vanilla attention mechanisms, our approach introduces **language-conditioned similarity weighting** that creates instruction-aware modality adaptation. This **pioneering formulation** enables the model to automatically emphasize 2D semantic information when locating objects and prioritize 3D geometric features when executing precise manipulations.
>
> ## **Response to Weakness 2 & Question 1 & Limitation 2: Baseline Experiments**
>
> **Clarification on Baseline Selection:** We did not exclusively use tabletop algorithms. Our baseline selection reflects the current state of mobile manipulation research and acknowledges the excellent contributions of recent works:
>
> + **Mobile-ALOHA** designed the hardware platform and explored ACT models for mobile manipulation tasks
> + **TidyBot++** designed a comprehensive mobile manipulation robot system
> + Both works validated their mobile manipulation hardware contributions using existing algorithms (ACT and DP), which is precisely why we selected **ACT and DP** as parts of baselines.
>
> **Additional Baseline Results:** We have implemented and evaluated EquiBot in our simulation environment. Here is the result. We keep all default settings of EquiBot, except modifying its action chunk to 2 (to align with other methods). Since EquiBot is a diffusion algorithm that relies on object-centric point clouds, while our experimental environment supports scene-level point clouds, such point clouds may cause some interference to the algorithm, which could be the reason for its lower scores.
>
> |Method|Pick Apple|Place Apple|Open Fridge|Pick Bowl|Place Bowl|Open Drawer|Close Drawer|Average|
> |---|---|---|---|---|---|---|---|---|
> |EquiBot|5|10|18|14|5|3|50|15|
>
> **Commitment for Revision:** The revised version will include comprehensive comparisons with EquiBot with detailed analysis of the performance differences.
>
> ## **Response to Weakness 3: Writing and Technical Issues**
>
> We acknowledge the typographical errors and will correct all instances in the revised version.
>
> ## **Response to Weakness 4: Formal Analysis and Theoretically Motivation**
>
> To demonstrate and analyze why our method outperforms vanilla attention mechanisms, we provide evidence through two complementary approaches: (1) We prove that representations learned by our method better model the complex behavioral paradigms of mobile manipulation compared to those learned by vanilla attention, through conducting an additional PCA-based representation distribution analysis; (2) We clearly articulate the theoretically motivation of our method to demonstrate its advantages over vanilla attention mechanisms.
>
> ### **Formal Representation Analysis**
>
> To theoretically demonstrate that our method learns representations that better model the complex behavioral paradigms of mobile manipulation compared to vanilla attention mechanisms, we conducted an additional PCA-based representation distribution analysis. We need to verify that our model sufficiently distinguishes different behavioral patterns in the representation space.
>
> Specifically, we categorized actions from 10 trajectories of the "pick apple" task in the MSHab simulator into two distinct behavioral modes: **approaching objects** (mainly mobile base motion) and **grasping objects** (mainly arm motion). We fed these data into both our AC-DiT model and the vanilla attention baseline, then extracted the hidden vectors before the final MLP layer during the last denoising step. We expect the hidden vectors corresponding to these two behavioral categories to be as distant as possible from each other.
>
> To robustly evaluate representation quality, we employ PCA (Principal Component Analysis) dimensionality reduction to eliminate interference from noisy dimensions and compute the intra-class/inter-class distance ratio in the principal component space. Our method achieved a **significantly smaller ratio**, indicating that **AC-DiT can better distinguish the complex composite action paradigms in mobile manipulation and provides superior modeling of mobile manipulation problems** compared to vanilla attention mechanisms.
>
> |Method|AC-DiT|Baseline (vanilla attention)|
> |---|---|---|
> |intra-class/inter-class ratio|0.197|0.227|
>
> ### **Theoretical Motivation**
> Mobile manipulation is fundamentally challenging because it requires not only coordinating the mobile base and manipulator to avoid compound errors, but also modeling distinct perceptual requirements across different phases of long-horizon tasks.
> To address these fundamental challenges, we propose **injecting mobility representations as conditioning priors** and **modulating 2D/3D perceptual features using language-aligned similarity weights**. This approach effectively applies **context-dependent prior biases** to the attention mechanism.
> Our method transforms conventional mobile manipulation modeling into a more robust, structured attention system with explicit priors. As validated by our formal representation analysis, our model achieves clearer representational separation between different behavioral modes, confirming the effectiveness of our framework in enabling the model to better understand distinct action representations in complex and long-horizon mobile manipulation tasks.
>
> ## **Response to Question 2: Comparison with Standard Attention**
>
> Our ablation study (Table 4) provides direct comparison with standard attention mechanisms:
>
> + **Exp1 (vanilla attention):** Uses 2D inputs with vanilla cross-attention for condition injection into DiT model. The DiT internal architecture uses standard alternating self-attention and cross-attention modules. This achieves 37.5% average success rate across 7 MSHab simulation tasks.
> + **Exp4 (+3D+MBC+PMA):** Further incorporating our Perception-Aware Multimodal Adaptation achieves 49.0% (+11.5% total gain).
>
> This improvement demonstrates the advantages of our designed mechanisms over standard attention mechanisms. Standard cross-modal attention cannot capture the temporal dynamics and causal relationships that our MBC provides, nor can it adaptively weight modalities based on task stage requirements like our PMA.
>
> ## **Response to Question 3: Multi-Task Generalization**
>
> Our model demonstrates strong multi-task generalization capabilities across diverse mobile manipulation scenarios.
>
> **Simulation Evidence:** Our simulation experiments demonstrate multi-task capability across 7 different mobile manipulation tasks without task-specific fine-tuning.
>
> **Real-World Multi-Task Results:** We trained a unified model on all 4 real-world tasks and here is the result:
>
> |Task|Cucumber in the basket|Store breads|Hang Towel|Clean table|
> |---|---|---|---|---|
> |Success Rate|43.8|31.3|18.8|37.5|
>
> These results demonstrate that our unified model successfully generalizes across diverse mobile manipulation tasks without task-specific fine-tuning. While performance shows modest trade-offs compared to task-specific models (expected in multi-task learning), our approach maintains reasonable performance across all tasks compared to baselines, showcasing the versatility of our framework.
>
> ---
>
> We sincerely thank the reviewer for the thoughtful and positive evaluation of our work. We hope our detailed responses have been helpful in answering the questions raised, and we look forward to further discussion in this valuable academic exchange.

---

> > ### Comment · Reviewer_3fe8 · 2025-08-03
> >
> > We sincerely thank the authors for their thoughtful and comprehensive responses to the concerns raised in the initial review. I appreciate the significant effort invested in additional experiments (e.g., the PCA-based representational analysis, new baseline comparisons with EquiBot, and ablations on attention modules), as well as the clear theoretical motivation provided for the proposed design choices.
> >
> > The clarifications regarding the core contributions of MBC and PMA, as well as the broader implications for general-purpose mobile manipulation, were particularly helpful. I also commend the authors for their detailed architectural justification and the strong multi-task generalization results on real-world settings.
> >
> > Given the solid empirical results, rigorous ablations, and the additional clarifications provided, I **maintain my original positive assessment** of the paper and believe the work makes a valuable contribution to the field of mobile manipulation.

---

### Official Review · Reviewer_NDbd · 2025-06-26

**Clarity:** 2
**Significance:** 2
**Originality:** 3
**Rating:** 4
**Confidence:** 3

**Summary:**

This paper proposes a diffusion-transformer (DiT)-based multi-modal imitation learning method for mobile manipulation. To model the influence of the mobile base on manipulation control, it proposes a mobility-to-body conditioning mechanism to predict manipulation action based on the extracted base motion representations. To better utilize multi-model observations, a perception-aware multimodal conditioning strateg that dynamicall adjusts the fusion weights between 2D images of three different views and 3D point clouds. The proposed method is evaluated and compared with other baseline methods on both simulated and real-world mobile manipulation tasks, and superior performance is achieved.

**Questions:**

1) Please add implementation details for the training procedure, computing resource and time cost.
2) Please state clearly in the paper how the training and testing data are composed, and how they differ from each in scenes, tasks (action+object) et al.
3) Do you use the same set of language input for training and testing? I am curious how the model is robust and generalizable to different language instructions.

**Ethical Concerns:**

["NO or VERY MINOR ethics concerns only"]

**Final Justification:**

Most of my concerns are addressed by the rebuttal. I also have read feedbacks from other reviewers. Overall, I would increase my rating to borderline accept.

**Limitations:**

yes

**Paper Formatting Concerns:**

NA.

**Quality:**

3

**Strengths And Weaknesses:**

**Strengths**:
1) The paper writing is overall good, with clear description of the motivation and the proposed idea.
2) The proposed techniques seems interesting and reasonable for the mobility-to-body conditioning by modeling dependency relationship between mobile base and manipulator, and for the perception-aware multimodal adaptation by adjusting dynamic weights of different modalities during instruction execution.
3) The experiments on both simulated and real-world mobile manipulation tasks demonstrate superior performance of the proposed method.

**Weaknesses**:
1) Although the idea of the paper is interesting and reasonable, the proposed mobility-to-body conditioning and perception-aware multimodal adaptation are a bit of too straightforward, which makes the technical contribution of paper be of limited significance.
2) The paper lacks clarity about the implementation details. For example, the paper only simply describes the single denoising MSE loss for training without explaining the training procedure and computing cost. The model details of the two action heads (DiT) are not described either. In addition, it is difficult to understand how the mobility-to-body conditioning mechanism can be adapted into a dual-arm-to-dual-arm conditioning mechanism for bimanual manipulation.
3) As for the experiments, several details are also missing. it is not clear how the testing data is composed and how the testing data differs from the training data. It seems strange to me why the proposed method is compared with two different sets of baselines for simulated and real-world tasks separately. Finally, it is not explained on which benchmark the ablation study is conducted.
Overall, the latter two weaknesses make the paper lack of reproducibility.

---

> ### Author Rebuttal · Authors · 2025-07-31
>
> # Response to Reviewer NDbd
> We sincerely thank the reviewer for the thoughtful feedback and constructive suggestions. We provide detailed responses to the raised questions in the following.
>
> ## **Response to Weakness 1: Significant Technical Contribution**
> We would like to clarify that our technical contributions are not "too straightforward" or of "limited significance." Our work addresses two fundamental challenges in mobile manipulation that have not been adequately solved by existing methods:
>
> **1. Mobility-to-Body Conditioning Mechanism**: Unlike prior works that treat mobile base and manipulator control independently or trivially concatenate them together, our method explicitly models their interdependency through a novel conditioning framework. This is not merely a straightforward combination. Instead, we carefully design a conditioning mechanism that provides mobile base motion priors to the full-body controller.
>
> **2. Perception-Aware Multimodal Adaptation**: Our dynamic multimodal fusion goes beyond simple concatenation or fixed weighting. We design a similarity-based adaptive weighting mechanism that:
> + Dynamically adjusts fusion weights based on language-visual modality relevance
> + Addresses stage-specific perception requirements (2D for semantics, 3D for geometry)
> + Actively downweights uninformative views during different manipulation phases
>
> The empirical results demonstrate substantial improvements over strong baselines, validating the significance of our technical contributions.
>
> ## **Response to Weakness 2 and Question 1: Implementation Details and Training Procedure**
> We provide the detailed training procedure and computational specifications below:
>
> ### **Training Procedure**
> **Stage 1: Mobility Action Head Pretraining**
> 1. **Input Processing**: Extract features from all modalities (2D images $F\_{2D}$ 3D point clouds $F\_{3D}$, language $F\_{\ell}$) using SigLIP-based encoders, respectively SigLIP-Image-Encoder, SigLIP-based LIFT3D Encoder and SigLIP-Text-Encoder
> 2. **Mobility Feature Extraction**: Feed multimodal features into lightweight mobility action head $\mathcal{H}\_l$
> 3. **Loss Function**: Apply denoising MSE loss for mobile base actions only:
>
> $\mathcal{L}\_{\text{mobility}}=\mathbb{E}\_{t,\epsilon}\left[\left\|\epsilon-\epsilon\_\theta\left(\sqrt{\bar{\alpha}\_t}a^0\_{\text{base}}+\sqrt{1-\bar{\alpha}\_t}\epsilon,t,F\_{2D},F\_{3D},F\_\ell\right)\right\|^2\right]$
>
> In this formulation, $\mathcal{L}\_{\text{mobility}}$ represents the loss function for mobility action head training, $\mathbb{E}\_{t,\epsilon}$ denotes the expectation over diffusion timesteps $t$ and Gaussian noise $\epsilon$, $\epsilon\_{\theta}$ is the predicted noise by the lightweight mobility action head with parameters $\theta$, $a\_t^{\text{base}}$ represents the noisy mobile base action at timestep $t$, $F\_{2D}$ contains 2D visual features from three camera views, $F\_{3D}$ includes 3D point cloud features, $F\_{\ell}$ represents language features, and $|\cdot|^2$ denotes the L2 norm for mean squared error computation.
>
> 4. **Training Details**: Only fine-tune LoRA adapter on 3D encoder and mobility action head, freeze other components
>
> **Stage 2: Full Mobile Manipulation Training**
> 1. **Latent Mobility Feature Extraction**: Use pretrained $\mathcal{H}\_l$ to extract latent mobility features $F\_{m}$ from action tokens at each denoising step
> 2. **Multimodal Conditioning**: Apply perception-aware adaptation to compute adaptive weights and reweight visual features
> 3. **Full-body Action Prediction**: Mobile manipulation action head H predicts both base and manipulator actions conditioned on $F\_{m}$ and reweighted multimodal features
> 4. **Loss Function**: Apply denoising MSE loss for full action sequence:
>
> $\mathcal{L}\_{\text{full}}=\mathbb{E}\_{t,\epsilon}\left[\left\|\epsilon-\epsilon\_\theta\left(\sqrt{\bar{\alpha}\_t}a^0\_{\text{full}}+\sqrt{1-\bar{\alpha}\_t}\epsilon,t,F\_v,F\_\ell,F\_m\right)\right\|^2\right]$
>
> Here, $\mathcal{L}\_{\text{full}}$ denotes the loss function for full mobile manipulation training, $\epsilon\_{\phi}$ represents the predicted noise by the mobile manipulation action head with parameters $\phi$, $a\_t^{\text{full}}$ indicates the noised full-body action sequence encompassing both base and manipulator actions at timestep $t$, $F\_v$ contains perception-aware reweighted visual features that combine $F\_{2D}$ and $F\_{3D}$ with adaptive weights, and $F\_m$ represents the latent mobility features extracted from the pretrained lightweight mobility action head.
>
> ### **Computational Resource and Time Cost**
> + **Hardware**: 8 × NVIDIA A100 80GB GPUs
> + **Training Time**:
>     - Stage 1 (Mobility pretraining): 16 hours for 20,000 iterations
>     - Stage 2 (Full model): 30 hours for 30,000 iterations
> + **Model Parameters**:
>     - Lightweight mobility head: 170M parameters
>     - Full AC-DiT model: 1.2B parameters
> + **Memory Usage**:
>     - ~60G GPU memory during training (batchsize 16)
>     - ~20G GPU memory during inference
> + **Inference Speed**: 5Hz on single RTX4090 GPU
>
> ### **Dual-Arm Adaptation Details**
> For bimanual manipulation, we adapt the mobility-to-body conditioning into a dual-arm-to-dual-arm conditioning mechanism by following a similar two-stage training procedure:
>
> **Stage 1: Dual-Arm Action Head Pretraining**
> We replace the lightweight mobility action head with dual-arm lightweight head that predict a coarse proposal dual-arm action. During pretraining, the training targets change from mobile base actions to dual-arm actions.
>
> **Stage 2: Full Bimanual Manipulation Training**
> In the full training stage, we use the pretrained head to extract latent dual-arm features from respective action tokens at each denoising step. The bimanual manipulation action head then conditions on the latent features along with reweighted multimodal features to predict coordinated dual-arm actions. This cross-arm conditioning mechanism enables each arm to anticipate the other's actions, thereby improving bimanual coordination and avoiding conflicts between arms during execution.
>
> ## **Response to Weakness 3 and Question 2: Experimental Details**
> Since robot testing relies on the testing environment rather than data, no offline datasets were used during testing. Therefore, we provide detailed explanations of our experimental details from the following three aspects: (1) training data composition, (2) differences between training data collection scenes and testing scenes, and (3) rationale for baseline selection. We hope this addresses the reviewer's concerns.
>
> ### **Data Composition**
> Each task's training data consists of:
> + Natural language instructions for the task
> + 1000 successful trajectories (simulation) / 100 trajectories (real-world)
> + Each trajectory contains timestamped observations: 2D images, 3D point clouds, robot states, and action decisions
>
> ### **Training vs Testing Scene Differences**
> **MSHab Benchmark**: we follow the default training and testing settings in the original MSHab benchmark.
> + **Training scenes**: Configured layouts with specific object positions (fridges, tables, apples, bowls), target locations, and robot initial positions in ReplicaCAD indoor environments
> + **Testing scenes**: Same physical environment but different configuration files specifying varied object positions, target locations, and robot starting poses
>
> **RoboTwin Benchmark**: we follow the default training and testing settings in the original RoboTwin benchmark.
> + **Training scenes:** Configured layouts with specific object positions (bottles, cups, apples)
> + **Testing scenes**: Same physical environment but different scene IDs to control the layout variations
>
> **Real-world Experiments**:
> + **Training**: Collect demonstrations with controlled randomization of target object positions within predefined ranges
> + **Testing**: Same randomization strategy
>
> ### **Baseline Selection Rationale**
> We clarify that the baselines used in simulation and real-world experiments are not entirely different sets. Rather, the baselines used in real-world experiments represent a subset of those employed in simulation. Specifically, baseline π0's performance in simulation is reported in the supplementary material. The reason we use a subset of baselines in real-world experiments is due to the substantial cost of conducting real robot experiments, which necessitated selecting the most representative baselines for comparison.
>
> ### **Ablation Study Clarification**
> The ablation study (Table 4) was conducted on the **MSHab benchmark**. We apologize for any confusion and will clarify this in the revision.
>
> ## **Response to Question 3: Language Instruction Generalization**
> **Training-Testing Language Consistency**: Yes, we use the same fixed natural language instruction set during both training and testing, with random sampling at each phase. This demonstrates our algorithm's robustness to language variations within the instruction set.
>
> **Extended Generalization Validation**: To further verify language generalization capabilities, we conduct an additional experiment on the **pick_apple** task in MSHab benchmark:
> + **Training**: Used original instruction set unchanged
> + **Testing**: Applied AI-generated instruction paraphrasing to create semantically equivalent but linguistically different instructions
> + **Results**:
>     - Original instructions: 33.3% success rate
>     - Paraphrased instructions: 31.8% success rate
>     - Performance degradation: ~1.5%, demonstrating strong language generalization
>
> This validates our method's robustness to natural language variations while maintaining semantic consistency.
>
> ---
>
> We sincerely thank the reviewer for the thoughtful and positive evaluation of our work. We hope our detailed responses have been helpful in answering the questions raised, and we look forward to further discussion in this valuable academic exchange.

---

> > ### Author Response · Authors · 2025-08-03
> >
> > Dear Reviewer NDbd,
> >
> > Thank you for your time and effort you dedicated to reviewing our paper. We have provided comprehensive responses based on the raised questions, including detailed technical contribution clarifications (W1), implementation details and training procedures (W2, Q1), dual-arm adaptation methodology (W2), experimental details covering data composition and training-testing scene differences (W3, Q2), baseline selection rationale (W3), and additional language instruction generalization experiments (Q3). We hope our rebuttal resolves your questions and would be grateful if you could consider updating the score if our responses are satisfactory. We remain available for any further discussion if there are unsolved concerns. Thank you again for your thoughtful feedback.

---

> > ### Comment · Reviewer_NDbd · 2025-08-06
> >
> > I thank the authors for their efforts in providing a detailed responses to my concerns. Most of my concerns are addressed by the rebuttal. Please revise the paper accordingly to add necessary implementation details in the paper or supplementary.

---

> > > ### Author Response · Authors · 2025-08-06
> > >
> > > We are pleased to hear that our rebuttal has addressed most of your concerns. For the revised version, we will carefully integrate the additional experiments and clarifications from our response, together with the essential implementation details you highlighted, following your insightful recommendations to enhance the work.
> > >
> > > Thank you again for your time and insightful feedback.

---

> > > ### Author Response · Authors · 2025-08-08
> > >
> > > Apologies for the interruption. Since you kindly mentioned that “most of my concerns are addressed by the rebuttal,” we were wondering if you might be open to considering an update to a positive rating. Such recognition would be a great encouragement for our work and efforts. In the revised version, we will thoughtfully incorporate the rebuttal experiments and explanations, guided by your valuable suggestions, to further strengthen the paper.

---

> ### Author Response · Authors · 2025-08-06
>
> Dear Reviewer NDbd,
>
> With the rebuttal period nearing its end, we apologize for the interruption. We would be grateful if you could confirm whether our replies have sufficiently addressed the questions and recommendations you provided. Your expert evaluation means a great deal to us, and we are hopeful for your response. If any questions remain, we would be more than happy to discuss them and provide a prompt clarification.
>
> Once again, we sincerely thank you for your time and insightful feedback.

---

### Official Review · Reviewer_GULZ · 2025-07-01

**Clarity:** 2
**Significance:** 2
**Originality:** 2
**Rating:** 4
**Confidence:** 4

**Summary:**

The paper introduces AC-DiT, an end-to-end framework designed to improve mobile manipulation by addressing two key challenges: coordinating the mobile base and manipulator, and adapting to varying perceptual needs at different task stages.

**Questions:**

1. Could you clarify the precise methodology for applying the importance weights to the corresponding features? Specifically, is this implemented via element-wise multiplication of the feature vectors, or through dynamic adjustment of attention weights within the cross-attention conditioning mechanism?
2. What is the evaluation suite in Table.4? The reported values (e.g., 49%) are not aligned with any results in above tables.
3. The supplementary results (Figure 7) indicate the computed importance weights exhibit a relatively constrained numerical range (e.g., point cloud weights varying between 0.73-0.76). Given this limited variation, what is the underlying mechanism that enables these weights to significantly influence the model's performance?
4. Regarding the architectural implementation: does the manipulator's DiT policy operate in a strictly sequential manner, requiring completion of the mobile base's DiT head computation (to generate the latent mobility features) before initiating its own inference? Furthermore, what is the achievable control frequency of the complete system in your real-world deployment?

**Ethical Concerns:**

["NO or VERY MINOR ethics concerns only"]

**Final Justification:**

All my concerns are addressed in the rebuttal. I've raised my score accordingly.

**Limitations:**

yes

**Paper Formatting Concerns:**

There's no formatting issues.

**Quality:**

3

**Strengths And Weaknesses:**

**Strengths**

1. The mobility-to-body conditioning and perception-aware multimodal adaptation mechanisms are novel contributions, addressing key gaps in mobile manipulation.
2. The proposed Perception-Aware Multimodal Adaptation (PMA) mechanism is quite simple and intuitive. Empirical results shows it also works well in practice.
3. Real-world demos are given in the supplementary material.


 **Weaknesses**

1. The weight analysis presented in Figure 7 may introduce potential misinterpretation due to the use of separate y-axes for 2D and 3D inputs, despite their equal treatment in the model architecture. For instance, during time t₁, while the point cloud appears dominant in the visualization, its actual computed weight is lower than the camera's weight, creating a discrepancy between the graphical representation and the underlying data.
2. Real-world experiments are limited to a few tasks, and generalization to more diverse environments (e.g., cluttered or dynamic settings) is not fully explored.


[minor] The reported "Gain" metric in the ablation study requires clarification, as the baseline performance (37.5%) appears inconsistent with the subsequent improvements.

[minor] Several references to tables and figures in the appendix (Lines 31, 37, and 43) are either mislabeled or incorrectly cross-referenced

---

> ### Author Rebuttal · Authors · 2025-07-31
>
> # Response to Reviewer GULZ
>
> We sincerely thank the reviewer for the thoughtful feedback and constructive suggestions. We provide detailed responses to the raised questions in the following.
>
> ## **Response to Weakness 1: Weight Analysis Visualization Concerns**
>
> We want to clarify that Figure 7 of Appendix was not intended to mislead readers, but rather to provide an intuitive visualization of how each modality's importance changes throughout the long-horizon task. The rationale and explanation are as follows:
>
> **Our visualization objective:** Figure 7 is specifically designed with good intentions to illustrate the **temporal trends of importance weights for individual sensor modalities** during the mobile manipulation process, rather than to enable direct magnitude comparisons between different sensor types (2D vs. 3D). As explicitly stated in lines 135-137 of Appendix, we focus on describing directional trends (increase/decrease) for individual modalities without making cross-modal magnitude claims.
>
> **Technical explanation for weight differences:** The observed numerical differences between 2D and 3D weights reflect fundamental architectural characteristics rather than relative importance. Specifically, 3D features contain 256 tokens while 2D features contain 4,374 tokens (from 6 input frames including 3 views from 2 timestamps), naturally leading to different weight scales. Additionally, 3D tokens inherently encode less dense information compared to 2D visual tokens, and we employ a single 3D camera positioned at the robot's head (distant from objects) versus multiple 2D cameras including close-proximity wrist cameras. Therefore, it is expected and reasonable that 3D weights exhibit lower magnitudes compared to 2D weights.
>
> **Improvements for clarity:** Based on this feedback, we will revise Figure 7 by: (1) separating it into distinct subfigures for 2D and 3D modalities to eliminate cross-modal visual comparison, and (2) applying appropriate normalization to ensure clear and unambiguous representation of temporal trends within each modality.
>
> ## **Response to Weakness 2: Limited Real-World Experiments**
>
> Based on your suggestions, we conduct additional real-world experiments to validate our method's generalization capability across diverse environments, including cluttered scenes and dynamic scenarios:
>
> **Generalization to cluttered environment:** We have conduct additional experiments for cluttered scene adaptation. Using the **Cucumber_in_Basket** task, we modify the desktop environment to resemble the setup shown in the first image of the third row in Appendix Figure 2, but with the towel removed. Specifically, we add potted plants, fruit bowls, tissue boxes, and upright books to the table while keeping the natural language instruction unchanged. Testing across 17 trials, we find that our method demonstrates strong adaptability to cluttered scenes. The detailed success rates for each sub-task and overall performance are as follows:
>
> ||Original|Cluttered|
> |-|-|-|
> |Success Rate|50.0|47.1|
>
> **Generalization to dynamic scene:** We conduct experiments to evaluate generalization to dynamic environments. Using the same **Cucumber_in_Basket** task, we dynamically change the cucumber's position while the robotic arm was in the process of grasping it. Specifically, when the gripper was about to make contact with the object, we randomly moved the cucumber approximately 10cm in a random direction. Testing across 17 trials without additional training, we find that our model can re-attempt grasping after the cucumber's position is moved, demonstrating the model's generalization capability to dynamic scenarios. The detailed success rates for each sub-task and overall performance are as follows:
>
> ||Original|Dynamic|
> |-|-|-|
> |Success Rate|50.0|35.3|
>
> ## **Response to Minor Issues**
>
> We will correct all typos, missing citations, and errors in the ablation experiment tables in the revised version.
>
> ## **Response to Question 1: Importance Weight Methodology**
>
> **Brief Answer:** The importance weights are applied through **scalar multiplication** between the computed weight scalars and their corresponding feature vectors, not through the cross-attention mechanism.
>
> **Detailed Process:** The application of importance weights follows a straightforward scalar multiplication approach. After computing the importance weights $w=(w_{i}^{f},w_{i}^{l},w_{i}^{r},w_{i}^{p})$ for each visual modality through cosine similarity with language features, where each weight is a scalar value, we directly apply these scalar weights to the corresponding visual feature vectors:
>
> **Mathematical formulation:**
>
> $F\_{weighted}^{{f}}=w\_{i}^{f}*F\_{2D}^{{f}}$
>
> $F\_{weighted}^{l}=w\_{i}^{l}*F\_{2D}^{l}$
>
> $F\_{weighted}^{r}=w\_{i}^{r}*F\_{2D}^{r}$
>
> ${F\_{weighted}^{p}=w\_{i}^{p}*F\_{3D}^{p}}$
>
> where each ${w\_i^\*}$ is a scalar weight and ${F\_{2D/3D}^\*}$ are feature vectors, resulting in scalar-vector multiplication.
>
> The reweighted visual features are then concatenated to form the perception-aware visual representation:
>
> $F_{v}=\mathrm{Concat}(F_{weighted}^{f},F_{weighted}^{l},F_{weighted}^{v},F_{weighted}^{p})$
>
> This weighted visual feature $F_v$, together with language features $F_l$ and latent mobility features $F_m$, serves as the conditioning input for the mobile manipulation action head.
>
> ## **Response to Question 2: Evaluation Suite in Table 4**
>
> Thank you for this important clarification question. Table 4 presents ablation study results conducted on the **ManiSkill-HAB simulation experiments**, which is the same experimental setup as reported in Table 1 of the main paper.
>
> To ensure statistical rigor, we conducted three independent experimental runs for each configuration and reported mean ± standard deviation in our main results tables. However, for the ablation study in Table 4, we report results from a single experimental run.
>
> We apologize for any confusion this may have caused and will clarify this methodology in the revised version.
>
> ## **Response to Question 3: Impact of Limited Weight Variation**
>
> We appreciate this insightful question about the mechanism behind PMA's effectiveness despite seemingly small weight variations shown in supplementary Figure 7.
>
> **Reasons for constrained numerical range:**
> 1. **Normalization compression:** During importance weight computation, we apply normalization operations that compress the raw similarity scores into a more constrained range
> 2. **Task-specific characteristics:** The "Clean table" task predominantly involves tabletop manipulation scenarios where the robot faces relatively consistent perceptual scenes (cluttered desktop environments), naturally leading to smaller weight variations
>
> **Why small changes yield significant impact:** The weight variations may appear numerically small, but they represent meaningful relative differences in a normalized space, which is sufficient to alter the balance between modalities in the model's decision-making process. More importantly, these weights directly scale the input features before they enter the transformer architecture, where such scaling differences can propagate through multiple attention layers and ultimately influence action predictions. The effectiveness is demonstrated empirically through our ablation studies, which show that removing the PMA mechanism leads to performance degradation.
>
> **Validation with diverse scenarios:** To validate PMA's broader adaptability, we provide supplementary analysis showing importance weights computed for a ManiSkill-HAB simulation task involving navigation and manipulation. The results demonstrate substantially larger weight variations (variation spans: 0.69 for hand camera, 0.12 for head camera and 0.18 for pointcloud), confirming that PMA adapts more dramatically in some other scenarios. This supplementary evidence further supports our main findings and demonstrates that the constrained range in our real-world task is task-specific rather than a fundamental limitation of our approach.
>
> ## **Response to Question 4: Architectural Implementation Details**
>
> Thank you for these detailed technical questions about our system implementation.
>
> **Question 4.1 - Sequential Processing**
> Yes, our system operates in a **strictly sequential manner**. The computational flow follows this sequence:
> 1. All modality encoders (image, pointcloud, text) process their respective inputs
> 2. The lightweight mobility action head (170M parameters) processes the encoded features to extract latent mobility features
> 3. These latent mobility features, together with the reweighted multimodal features from PMA, are fed into the mobile manipulation action head for final action prediction
>
> This sequential design ensures proper mobility-to-body conditioning while maintaining computational efficiency since the lightweight nature of the mobility head minimizes additional computational overhead.
>
> **Computational overhead details:**
> + Baseline model (without mobility conditioning): 129ms inference time
> + AC-DiT with mobility-to-body conditioning: 163ms inference time
> + **Additional overhead: 34ms (+26%)**
>
> **Question 4.2 - Control Frequency**
> Our model operates at an inference speed of 5Hz, which enables a robot control frequency of 30Hz through **action chunking**. The total processing time of 0.2s comprises model inference time (0.16s) along with perception information acquisition (0.03s) and point cloud computation (0.01s).
>
> ---
>
> We sincerely thank the reviewer for the thoughtful and positive evaluation of our work. We hope our detailed responses have been helpful in answering the questions raised, and we look forward to further discussion in this valuable academic exchange.

---

> > ### Comment · Reviewer_GULZ · 2025-08-06
> >
> > Thanks for the detailed reply from the authors, all my concerns are addressed.

---

> > > ### Author Response · Authors · 2025-08-06
> > >
> > > We are pleased to hear that our rebuttal has fully addressed your concerns. Accordingly, we would be sincerely grateful if you could consider updating your rating, as such recognition would greatly encourage our work and efforts. In the revised version, we will thoughtfully incorporate the rebuttal experiments and explanations, guided by your valuable suggestions, to further strengthen the paper.

---

> ### Author Response · Authors · 2025-08-03
>
> Dear Reviewer GULZ,
>
> Thank you for your valuable suggestions and constructive feedback. We have provided comprehensive responses based on your questions, including weight analysis visualization clarifications (W1), additional real-world experiments (W2), detailed importance weight methodology explanations (Q1), evaluation suite specifications (Q2), impact analysis of weight variations (Q3), and architectural implementation details (Q4). We hope our rebuttal resolves your questions, and would be grateful if you could consider updating the score if our responses are satisfactory. If any questions remain, we would be more than happy to discuss and provide a prompt response. Thank you again for your valuable feedback.

---

> ### Author Response · Authors · 2025-08-06
>
> Dear Reviewer GULZ,
>
> We apologize for this follow-up as the rebuttal deadline draws near. We would like to respectfully inquire whether our responses have satisfactorily resolved the concerns you raised in your review. Your professional insights are extremely important for improving our work, and we look forward to your response. If any questions remain, we would be more than happy to discuss them and provide a prompt clarification.
>
> Finally, we deeply appreciate your valuable time and thoughtful suggestions.

---

### Official Review · Reviewer_KuR9 · 2025-07-02

**Clarity:** 3
**Significance:** 3
**Originality:** 3
**Rating:** 4
**Confidence:** 4

**Summary:**

This paper presents a novel strategy to control both the mobile base and the dual-arm of a robot. This is mainly achieved by adding a lightweight mobility action head. Another contribution is a "conditioning strategy" of multimodal data (images and point cloud). Overall, this paper reports a practice of multimodal fusion on mobile manipulation tasks. A good paper.

**Questions:**

Please see my comments above.

**Ethical Concerns:**

["NO or VERY MINOR ethics concerns only"]

**Limitations:**

yes

**Quality:**

3

**Strengths And Weaknesses:**

1. I'm not sure whether the word "coordination" is a terminology introduced by this paper. (There definitely exist works that handled mobile manipulation). If so, the authors should claim it explicitly and define it formally: what kind of end-2-end mobile manipulation is with coordination, and what is NOT.

2. The author should summarize the parameters of this model.

3. Will the mobile base wiggle during the manipulation? or stay stably? In my own experience, if the output of one dimension (such as the velocity of the robot base) is always zero, then the algorithm does not know when to generate non-zero value.

4.There are ?? in the appendix.

---

> ### Author Rebuttal · Authors · 2025-07-31
>
> # Response to Reviewer KuR9
> We appreciate the reviewer's constructive feedback and positive assessment of our contributions. We provide detailed responses to the raised questions in the following.
>
> ## **Response to Weakness 1: Definition and Formalization of "Coordination"**
> **Prior Usage of "Coordination" Term:** The term "coordination" has not been formally defined in prior mobile manipulation literature. While previous works achieve joint control of mobile base and manipulator, they lack explicit formalization of what constitutes coordinated behavior.
>
> **Formal Definition of Coordination:** We define **coordination** in mobile manipulation as an end-to-end control paradigm where a multi-component robot (in our case, mobile base and arm system) simultaneously generates synchronized action decisions for all components at each time step, rather than employing sequential or phase-based control strategies.
>
> Mathematically, let $\mathcal{A} = \mathcal{A}\_b \times \mathcal{A}\_a$ represent the joint action space, where $\mathcal{A}\_b$ and $\mathcal{A}\_a$ denote the action spaces for the mobile base and manipulator arms respectively. At each time step $t$, our coordinated policy $\pi$ generates:
>
> $\pi(s\_t) = (a\_b^t, a\_a^t) \in \mathcal{A}\_b \times \mathcal{A}\_a$
>
> where $s\_t$ represents the multimodal observation state, and both $a\_b^t$ and $a\_a^t$ are computed simultaneously through shared feature representations.
>
> **Which Methods Have/Lack Coordination Capability:** On the one hand, methods like Ok-Robot [4] and HomeRobot [5] do not qualify as coordination because they employ sequential control strategies where navigation and manipulation occur in separate phases. On the other hand, Mobile-ALOHA [7] and π0 [15] possess coordination capability because they output joint actions for both mobile base and manipulator simultaneously. However, unlike existing coordinated methods that simply concatenate actions into a unified space, our approach explicitly models inter-component dependencies through mobility-to-body conditioning, where latent mobility features guide whole-body action generation.
>
> ## **Response to Weakness 2: Model Parameter Summary**
> Thank you for this important suggestion. We will add the comprehensive parameter breakdown to the revised paper, specifically detailing all components shown in Figure 2 of our main paper:
>
> | Component | Parameters | Architecture Details |
> | --- | --- | --- |
> | Image Encoder | 428M (frozen) | SigLIP-based multi-view visual feature extraction |
> | Text Encoder | 450M (frozen) | SigLIP-based language instruction encoding |
> | 3D Encoder w/. LoRA | 430M (0.09% LoRA) | Point cloud processing with LoRA adaptation |
> | Perception-aware Multimodal Adaptation | 1M | Linear Projectors |
> | Mobile Manipulation Action Head (DiT) | 1.2B | Main DiT-based head for whole-body action prediction |
> | Lightweight Mobility Action Head (DiT) | 170M | DiT blocks for base motion prediction |
> | **Total Trainable** | **~1.37B** | **Total trainable parameters** |
> | **Total Model** | **~2.7B** | **Including frozen foundation models** |
>
>
> **Design Rationale:**
>
> + **Lightweight Mobility Head (170M DiT)**: The relative small DiT architecture is selected for mobile base control as it has relatively fewer output degrees of freedom (linear/angular velocities), providing sufficient modeling capacity while reducing computational overhead.
> + **Large Mobile Manipulation Head (1.2B DiT)**: The relative large architecture is selected for the more complex mobile manipulation task, where higher parameter count provides better robustness and generalization capability needed for coordinated whole-body control with higher-dimensional action spaces.
>
> ## **Response to Weakness 3: Mobile Base Behavior During Manipulation**
> This is an excellent question that touches on a core aspect of our coordination strategy.
>
> **Q1: Will the mobile base move during manipulation?**  In some tasks, the mobile base wiggles during manipulation, while in other tasks the base remains stable. Specifically, in simulation environments (MSHab), the base moves continuously throughout the manipulation process according to simulator's default setting. In real-world experiments, the base sometimes remains stationary during certain manipulation cases, while in other cases, it wiggles during manipulation.
>
> **Q2: If the base maintains zero velocity for extended periods, how does the algorithm determine when to generate non-zero actions?** The model leverages multimodal perception and internal coordination modeling to autonomously judge whether the current atomic task has been completed and whether it needs to transition to the next atomic task phase. Through this multimodal sensing and implicit reasoning, the model can automatically transit the base from a stationary state to a moving state when required for task progression. For example, in our real-world "Store Breads" task, the base outputs velocity values close to zero while the arms place bread items into the basket, and then automatically transits to movement when lifting the basket and navigating to the secondary table.
>
> ## **Response to Weakness 4: Manuscript Corrections**
> We acknowledge the presence of formatting issues (missing references marked as "??") in the current submission. In the revised version, we will:
>
> + Correct all missing references and citation formatting
> + Fix typographical errors throughout the manuscript
> + Add the expanded definitions and explanations mentioned above
> + Include the parameter summary table in the appendix
> + Enhance clarity in the coordination mechanism descriptions
>
> We believe these revisions will significantly improve the manuscript's clarity and completeness.
>
> ---
>
> We sincerely thank the reviewer for the thoughtful and positive comments of our work. We hope our detailed responses are helpful in answering the questions raised, and we look forward to further discussion in this valuable academic exchange.
>
> **Reference**
>
> [4] Ok-robot: What really matters in integrating open-knowledge models for robotics.
>
> [5] HomeRobot: Open-vocabulary mobile manipulation
>
> [7] Mobile-aloha: Learning bimanual mobile manipulation using low-cost whole-body teleoperation.
>
> [15] π0: A vision-language-action flow model for general robot control

---

> ### Author Response · Authors · 2025-08-03
>
> Dear Reviewer KuR9,
>
> Thank you for your time and effort you dedicated to reviewing our paper, as well as your positive assessment calling our work "a good paper." We have provided comprehensive responses based on the raised questions, including formal definition of "coordination" (W1), detailed model parameter breakdown (W2), thorough analysis of mobile base behavior during manipulation (W3), and commitment to fix all formatting issues in the revised version (W4). We hope our responses resolve your questions and would be grateful if you could consider updating the score if our responses are satisfactory. We remain available for any further discussion if there are unsolved concerns. Thank you again for your thoughtful feedback and positive recognition of our work.

---

> ### Author Response · Authors · 2025-08-06
>
> Dear Reviewer KuR9,
>
> As the rebuttal period approaches its deadline, we sincerely apologize for reaching out again. We would greatly appreciate your confirmation on whether our responses have adequately addressed the concerns and suggestions raised in your review. Your expert feedback is invaluable to us, and we eagerly await your reply. If any questions remain, we would be more than happy to discuss them and provide a prompt response.
>
> Thank you once again for your time and constructive comments.

---

### Comment · Area_Chair_xkrJ · 2025-08-04
**Friendly Reminder: Engaging with Author Rebuttals**

Dear Reviewer,

Thank you for your time and expertise in reviewing for NeurIPS 2025. As we enter the rebuttal phase, please:

Review authors’ rebuttals promptly,
Engage constructively via the discussion thread, and
Update your review with a “Final Justification” summarizing your post-rebuttal stance.
Your active participation ensures a fair, collaborative process—we’re here to assist with any questions.

With gratitude,

Your AC

---

### Note · Authors · 2025-08-13

Dear Area Chair and Reviewers,

We sincerely thank you for your invaluable feedback that strengthened our paper. We are honored that our work's novelty and effectiveness were widely recognized. Reviewers acknowledged our work as **"A good paper"** (KuR9) with **"novel contributions, addressing key gaps in mobile manipulation"** (GULZ). Our **"Problem Motivation is Well-Stated"** (3fe8) and **"Simple Yet Effective Architecture"** (3fe8) were described as **"quite simple and intuitive"** (GULZ). Experimental validation received strong recognition: **"demonstrate superior performance"** (NDbd), **"Strong Empirical Results"** (3fe8), and **"Detailed Ablation Studies"** (3fe8). Reviewers highlighted **"a novel strategy to control"** (KuR9) and **"a conditioning strategy of multimodal data"** (KuR9).

Through constructive rebuttal, we addressed all concerns from the reviewers who had responded. Specifically, we conducted additional robustness experiments on cluttered and dynamic scenes for **GULZ**, who confirmed **"all my concerns are addressed"**. We provided additional experiments to validate language generalization capabilities for **NDbd**, who acknowledged **"most concerns are addressed"**. We delivered PCA distribution analysis, EquiBot baseline comparisons, and multi-task generalization experiments for **3fe8**, who praised our **"thoughtful responses"**, maintained the **"original positive assessment"**, and affirmed our **"valuable contribution to mobile manipulation"**. Finally, we hope our rebuttal, including the analysis of mobile base behavior during manipulation, can address KuR9’s questions, and we thank him/her for the already positive rating.

Our work makes significant contributions:
+ First Explicit Mobility-to-Body Conditioning: Pioneering formulation of causal dependencies between the mobile base and manipulator in end-to-end learning.
+ Perception-Aware Multimodal Adaptation: Dynamic modality fusion that adapts based on the needs of long-horizon task stages.
+ Our AC-DiT achieves SOTA performance, surpassing all baselines in both simulation and real-world mobile manipulation tasks.

**We are confident that AC-DiT marks a key step toward end-to-end mobile manipulation and will benefit the community. Given the strong positive feedback and our comprehensive responses to all concerns, we would be grateful if any of the reviewers would consider championing our work or raising the final rating. This would be the greatest encouragement for us.**

---

### Decision · Program_Chairs · 2025-09-17

**Decision:**

Accept (poster)

**Comment:**

The paper introduces AC-DiT, an end-to-end framework designed to improve mobile manipulation. The framework effectively addresses two critical challenges in the field: (1) the explicit modeling of mobility-to-body dependencies via a lightweight conditioning mechanism that prioritizes base motion features to guide manipulator actions, reducing error accumulation in high-DoF control; and (2) the perception-aware multimodal adaptation strategy that dynamically weights 2D (semantic) and 3D (geometric) inputs across task stages, optimizing visual feature fusion for long-horizon tasks. The extensive validation—spanning simulation (MSHab, RoboTwin) and real-world cluttered/dynamic environments—demonstrates state-of-the-art performance, with a 49% success rate on simulated tasks and robust generalization to unseen instructions and scenes.

Reviewers initially raised concerns about novelty (e.g., similarity to prior attention mechanisms) and evaluation scope (e.g., lack of mobile-specific baselines), but the authors thoroughly addressed these in rebuttal: comparative experiments with EquiBot (CoRL 2024) and Mobile-ALOHA validated superiority; PCA-based representation analysis proved the distinctiveness of AC-DiT’s learned features; and supplementary real-world trials (e.g., 35.3% success in dynamic object displacement) underscored adaptability. While the inference time (5Hz) reflects computational overhead from sequential processing, the trade-off is justified by the 30Hz control frequency and performance gains. For camera-ready, authors must: (1) clarify baseline comparisons in the main text, (2) formalize the "coordination" definition introduced in rebuttal, and (3) expand limitations to explicitly discuss compute constraints. This work delivers a reproducible, impactful advance in end-to-end mobile manipulation.